# Distractor-free Generalizable 3D Gaussian Splatting

**Yanqi Bao**
Nanjing University
Jiangsu, Nanjing, China
yanqibao1997@gmail.com

Jing Liao*
City University of Hong Kong
Hong Kong, China
jingliao@cityu.edu.hk

Jing Huo*
Nanjing University
Jiangsu, Nanjing, China
huojing@nju.edu.cn

Yang Gao
Nanjing University
Jiangsu, Nanjing, China
gaoy@nju.edu.cn

## ABSTRACT

We present DGGS, a novel framework that addresses the previously unexplored challenge: **Distractor-free Generalizable 3D Gaussian Splatting** (3DGS). Previous generalizable 3DGS works are often limited to static scenes, struggling to mitigate distractor impacts in training and inference phases, which leads to training instability and inference artifacts. To address this new challenge, we propose a distractor-free generalizable training paradigm and corresponding inference framework, which can be directly integrated into existing Generalizable 3DGS frameworks. Specifically, in our training paradigm, DGGS proposes a feed-forward mask prediction and refinement module based on the 3D consistency of references and semantic prior, effectively eliminating the impact of distractor on training loss. Based on these masks, we combat distractor-induced artifacts and holes at inference time through a novel two-stage inference framework for reference scoring and re-selection, complemented by a distractor pruning mechanism that further removes residual distractor 3DGS-primitive influences. Extensive feed-forward experiments on the real and our synthetic data show DGGS's reconstruction capability when dealing with novel distractor scenes. Moreover, our feed-forward mask prediction even achieves an accuracy superior to scene-specific Distractor-free methods.

## 1 INTRODUCTION

The widespread availability of mobile devices presents unprecedented opportunities for 3D reconstruction, fostering demand for feed-forward 3D synthesis capabilities from casually captured images or video sequences (referred to as references). Recent approaches introduce generalizable 3D representations to address this challenge, eliminating per-scene optimization requirements, with 3D Gaussian Splatting (3DGS) demonstrating particular promise due to its efficiency (Charatan et al., 2024; Liu et al., 2025; Chen et al., 2024b; Zhang et al., 2024a). In pursuit of scene-agnostic inference from references to 3DGS, existing methods project reference features onto 3D space to predict 3DGS attributes and simulate the complete pipeline: input **references**, infer **3DGS**, and **render** novel query views, within each training step. This process utilizes selected reference-query pairs for training and optimizes the model to learn the reference-query 3D consistency through query rendering losses.

While promising, this paradigm faces two major challenges in real-world, unconstrained capture scenarios due to the presence of distractor (e.g., transient objects such as vehicles or pedestrians). First, during training, real-world data often contains distractor that disrupt 3D consistency, limiting training to confined static scenes. Second, during inference, distractor in the reference images cannot be properly projected into 3D space, resulting in unwanted artifacts in the reconstructed 3D scene.

To tackle these issues, we propose Distractor-free Generalizable 3D Gaussian Splatting (DGGS), a novel framework that enhances training stability when training generalizable 3DGS models under

---

*Corresponding author

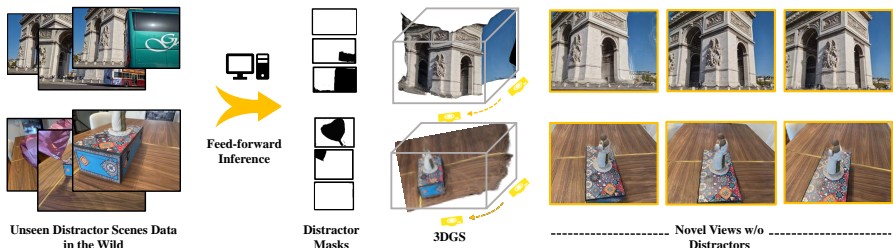

Figure 1: **Overview of Our Task.** *Distractors are unwanted transient objects in static scene reconstruction, such as buses, balloons, or anything.* DGGS enables feed-forward 3DGS reconstruction from limited distractor data while inferring corresponding distractor masks without extra supervision.

distractor-data and mitigates distractor-induced artifacts during the inference process. This framework builds on two key components: **a Distractor-free Generalizable Training paradigm** and **a Distractor-free Generalizable Inference framework**. The core idea behind them lies in the discussion about how to predict distractor masks in a feed-forward manner and use them for training and inference. Unlike existing scene-specific distractor-free masking methods that require sufficient input and iterative optimization (Chen et al., 2024a; Ungermann et al., 2024; Sabour et al., 2024), our method takes advantage of the inherent **3D consistency across references** to infer distractor masks in each training iteration. These masks are then applied to exclude distractor regions from reconstruction loss during training, improving stability, and are further used during inference to prioritize cleaner references and suppress artifacts.

Specifically, in **Distractor-free Generalizable Training paradigm**, we design a **Reference-based Mask Prediction** and a **Mask Refinement** module for generalizable distractor masking, which is based on a core observation, *re-rendered reference non-distractor areas from 3DGS (inferred by references) are generally accurate and robust.* Therefore, given an initialized mask, we use these non-distractor areas in reference as guidance to filter out misclassified distractor regions in query. This process relies on the static geometric multi-view consistency of non-distractor areas. For higher mask accuracy, after decoupling the filtered masks into distractor and disparity error areas, our mask refinement module incorporates pre-trained segmentation results to fill distractor regions and designs a reference-based auxiliary loss to additionally supervise the unnoticed occluded regions in query view. Based on these masks, we propose a two-stage **Distractor-free Generalizable Inference framework** for mitigating holes and artifacts at inference time. In the first stage, we introduce more candidate references and design a **Reference Scoring** mechanism to score candidator based on the predicted distractor masks. These predicted scores guide the references selection with minimal distractors and disparity for fine reconstruction in the second stage. To further mitigate ghosting artifacts from residual distractor in the second stage, we introduce a **Distractor Pruning** strategy that eliminates distractor-associated 3D gaussian primitives.

Overall, we address a new task of *Distractor-free Generalizable 3DGS* as in Fig. 1, and this is, to our knowledge, the first work to explore this problem. For this objective, we present **DGGS**, a framework designed to alleviate distractor-related adverse effects during the training and inference phases of generalizable 3DGS. Extensive feed-forward experiments on real distractor data have shown that our approach successfully enhances training robustness and improves artifacts and holes during inference while expanding cross-scene (outdoor scenes training and indoor scenes inference) generalizability in conventional scene-specific distractor-free models (Chen et al., 2024a; Sabour et al., 2023; Ren et al., 2024; Sabour et al., 2024). Beyond real data, we also construct some synthetic distractor scenes based on Re10K and ACID datasets for further verification. Furthermore, our reference-based training paradigm achieves better generalizable distractor masking *without any mask supervision*, even outperforming scene-specific training distractor-free works(Chen et al., 2024a).

## 2 RELATED WORKS

### 2.1 GENERALIZABLE 3D RECONSTRUCTION

Contemporary advances in generalizable 3D reconstruction seek to establish scene-agnostic representations, building upon early explorations in Neural Radiance Fields (Mildenhall et al., 2021; Wang et al., 2021; 2022; Liu et al., 2022; Bao et al., 2023a). However, these methods face significant bottlenecks due to their lack of explicit representations and rendering inefficiencies. The advent

of 3D Gaussian Splatting (Kerbl et al., 2023), an explicit representation optimized for efficient rendering, has sparked renewed interest in the field. Existing works involve inferring Gaussian primitive attributes from references directly and rendering in novel views. Analogous to NeRF-based approaches, 3DGS-related methods emphasize spatial comprehension from references, particularly focusing on depth estimation (Charatan et al., 2024; Chen et al., 2024b; Liu et al., 2025; Zhang et al., 2024a; Liang et al., 2023). Subsequently, ReconX (Liu et al., 2024) and G3R (Chen et al., 2025) enhance reconstruction quality through the integration of additional video diffusion models and supplementary sensor inputs. The inherent reliance on high-quality references, however, makes generalizable reconstruction particularly susceptible to **distractor**, a persistent challenge in real-world applications. In this study, we discuss Distractor-free Generalizable 3DGS, an unexplored topic.

## 2.2 SCENE-SPECIFIC DISTRACTOR-FREE RECONSTRUCTION

It focuses on accurately reconstructing a *specific* static scene while mitigating the impact of distractor (Ren et al., 2024) (or transient objects (Sabour et al., 2023)). As a pioneering work, NeRF-W (Martin-Brualla et al., 2021) introduces additional embeddings to represent and eliminate transient objects in unstructured photo collections. Following a similar setting, subsequent works focus on mitigating the impact of distractor *at the image level*, which can generally be categorized into knowledge-based and heuristics-based methods.
**Knowledge-based methods** predict distractor using external knowledge sources. Among them, pre-trained features from ResNet (Zhang et al., 2024b; Xu et al., 2024), diffusion models (Sabour et al., 2024), and DINO (Ren et al., 2024; Kulhanek et al., 2024) guide visibility map prediction, effectively weighting reconstruction loss. Recent works (Chen et al., 2024a; Otonari et al., 2024; Nguyen et al., 2024) directly employ state-of-the-art segmentation models such as SAM (Kirillov et al., 2023) or Entity Segmentation (Qi et al., 2022) to establish clear distractor boundaries. Although these approaches demonstrate certain improvements (Martin-Brualla et al., 2021; Chen et al., 2022; Lee et al., 2023) with additional priors, they struggle to differentiate the distractor from static target scenes (Chen et al., 2024a; Otonari et al., 2024). **Heuristics-based approaches** employ handcrafted statistical metrics to distinguish distractor, emphasizing robustness and uncertainty analysis (Sabour et al., 2023; Goli et al., 2024; Ungermann et al., 2024). These methods exploit the observation that regions containing distractor typically manifest optimization inconsistencies. Therefore, they seek to predict outliers and mitigate their impact in residual losses. Regrettably, these approaches suffer from significant scene-specific data dependencies and frequently confound distractor with inherently challenging reconstruction regions, limiting their effectiveness in generalizable contexts.

Recently, there has been increasing advocacy for integrating the two methods (Otonari et al., 2024; Chen et al., 2024a). Entity-NeRF (Otonari et al., 2024) integrates an existing Entity Segmentation (Qi et al., 2022) and extra entity classifier to determine distractor among entities by analyzing the rank of residual loss. Similarly, NeRF-HuGS (Chen et al., 2024a) integrates pre-defined Colmap and Nerfacto (Tancik et al., 2023) for capturing high and low-frequency features of static targets, while using SAM (Kirillov et al., 2023) to predict clear distractor masks. However, in our settings, acquiring additional entity classifiers or employing pre-defined scene-level knowledge such as Colmap and Nerfacto is nearly impossible, and residual loss becomes unreliable compared to single-scene optimization due to the absence of iteratively refined representation. Moreover, with limited references in unseen scenes, despite obtaining distractor masks, traditional Distractor-free methods struggle to handle occluded regions and artifacts. Therefore, we present a novel **Distractor-free Generalizable** framework that jointly addresses distractor effects in training and inference phases.

## 3 PRELIMINARIES

### 3.1 3D GAUSSIAN SPLATTING

3DGS (Kerbl et al., 2023; Bao et al., 2025) $\mathcal{G}$ represents 3D scenes by splatting numerous anisotropic gaussian primitives. Each gaussian primitive is characterized by a set of attributes $\mathbb{A}$, including position $p$, opacity $\alpha$, covariance matrix $\Sigma$, and spherical harmonic coefficients for color $\hat{c}$. To ensure positive semi-definiteness, the covariance matrix $\Sigma$ is decomposed into a scaling matrix $\mathbf{S}$ and a rotation matrix $\mathbf{R}$, such that $\Sigma = \mathbf{R}\mathbf{S}\mathbf{S}^\top\mathbf{R}^\top$. Consequently, the color value after splatting on view $\mathbf{P}$ is:

$$\hat{C} = \mathcal{G}(\mathbf{P}) = \sum_{m \in M} \hat{c}_m \alpha_m \prod_{j=1}^{m-1} (1 - \alpha_j), \tag{1}$$

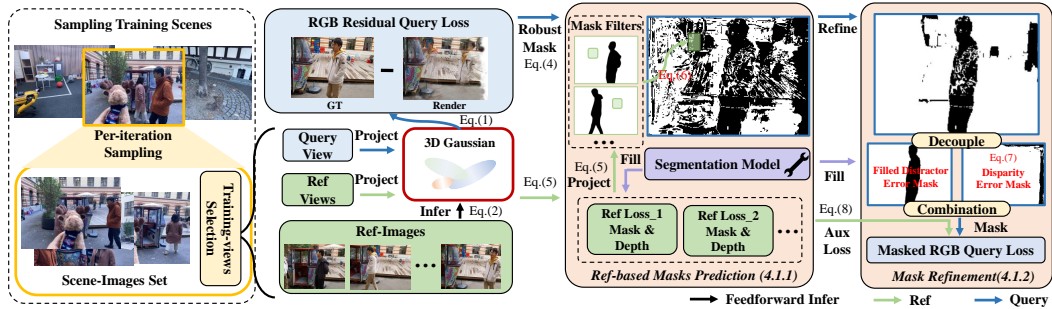

Figure 2: **Distractor-free Generalizable Training.** Based on the sampled reference-query pairs, DGGS first predicts 3DGS attributes and a fundamental robust mask $\mathcal{M}_{Rob}$. The **Reference-based Mask Prediction** module then filters this mask, which is further refined through the **Mask Refinement** module. The entire process is supervised through masked query loss and auxiliary loss.

where $\hat{c}_m$ and $\alpha_m$ are derived from the covariance matrix $\mathbf{\Sigma}_m$ of the $m$-th projected 2D Gaussian, as well as the corresponding spherical harmonic coefficients and opacity, respectively.

## 3.2 GENERALIZABLE 3DGS

Generalizable 3DGS presents a novel paradigm that directly infers 3DGS $\mathcal{G}$ attributes from references, circumventing the computational overhead of scene-specific optimization. During each training iteration, existing works optimize model parameters $\boldsymbol{\theta}$ by randomly sampling paired references $\{\mathbf{I}_i\}_{i=1}^{N}$ and query image $\mathbf{I}_T$ as inputs and groundtruth under random sampled scenes. Specifically,

$$\mathcal{G} = \text{Decoder}\left(\mathcal{F}\left(\text{Encoder}\left(\{\mathbf{I}_i\}_{i=1}^{N}\right), \{\mathbf{P}_i\}_{i=1}^{N}\right)\right), \quad \arg\min_{\boldsymbol{\theta}} \|\mathbf{I}_T - \mathcal{G}(\mathbf{P}_T)\|_2^2, \quad (2)$$

where $\{\mathbf{P}_i\}_{i=1}^{N}$ and $\mathbf{P}_T$ are the camera extrinsics of the reference and query images, and $N$ denotes the number of references. The $\mathcal{F}$ denotes the process of projecting encoded 2D features into 3D space and refining (Chen et al., 2024b; Charatan et al., 2024). The 3D features are then decoded into the corresponding 3DGS attributes and rendered into query views, through which the network can be optimized by the query MSE loss.

## 3.3 ROBUST MASKS FOR 3D RECONSTRUCTION

Unlike conventional controlled static environments, our in-the-wild scenarios contain not only static elements but also distractor (Sabour et al., 2023), making it difficult to maintain 3D geometric consistency and training stability. Building on prior Scene-specific Distractor-free reconstruction research (Sabour et al., 2023; 2024; Ren et al., 2024; Chen et al., 2024a), we integrate the mask-based robust optimization loss in our pipeline (i.e. 'MVSplat+ *' in experiment, where * represents different mask prediction methods) that can predict and filter out distractor (outliers) in training process. Hence, Eq. 2 is modified, where $\odot$ denotes pixel-wise multiplication:

$$\arg\min_{\boldsymbol{\theta}} \mathcal{M}_{Rob} \odot \|\mathbf{I}_T - \mathcal{G}(\mathbf{P}_T)\|_2^2. \quad (3)$$

Here, $\mathcal{M}_{Rob}$ represents the outlier masks on query view, where distractor is set to zero, while target static regions are set to one. Our work introduces a simple heuristic method (Sabour et al., 2023) as the foundation. The hyperparameters $\rho_1$ and $\rho_2$ are fixed across all scenes.

$$\mathcal{M}_{Rob} = \mathbb{1}\left\{\mathcal{C}\left(\mathbb{1}\left\{\|\mathbf{I}_T - \mathcal{G}(\mathbf{P}_T)\|_2 < \rho_1\right\}\right) > \rho_2\right\}, \quad (4)$$

where $\mathcal{C}$ represents the kernel operator and the $\rho_1$ as well as $\rho_2$ remain consistent with (Sabour et al., 2023). Despite various mask refinements in follow-up studies (Otonari et al., 2024; Chen et al., 2024a; Sabour et al., 2024), their heavy dependence on loss $\|\mathbf{I}_T - \mathcal{G}(\mathbf{P}_T)\|_2$ leads to extensive misclassification of difficult-to-feed-forward-inference parts as distractor regions, as show in Fig. 3 and Fig. 6, which is addressed in subsequent sections.

## 4 METHOD

Given sufficient training reference-query pairs, the presence of distractor in either $\{\mathbf{I}_i\}_{i=1}^{N}$ or $\mathbf{I}_T$ (or both) affects the 3D consistency relied upon by generalizable models. Therefore, we aim to design a **Distractor-free Generalizable** Training paradigm in Sec. 4.1 and a Inference framework in Sec. 4.2.

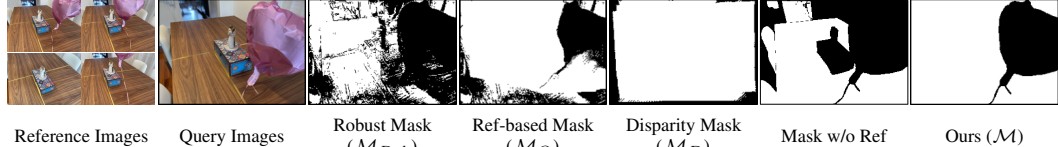

| Reference Images | Query Images | Robust Mask $(\mathcal{M}_{Rob})$ | Ref-based Mask $(\mathcal{M}_Q)$ | Disparity Mask $(\mathcal{M}_D)$ | Mask w/o Ref | Ours ($\mathcal{M}$) |

Figure 3: **The Mask Evolution in Sec. 4.1.** $\mathcal{M}_Q$ is obtained by filtering $\mathcal{M}_{Rob}$ from the references non-distractor regions, which is then filled by decoupling $\mathcal{M}_D$ and using segmentation results to get final $\mathcal{M}$ as Eq. 4 6 7. Without references filter, target regions are often misidentified as distractor.

## 4.1 DISTRACTOR-FREE GENERALIZABLE TRAINING

In this training paradigm, we propose a **Reference-based Mask Prediction** in Sec.4.1.1 and a **Mask Refinement** module in Sec.4.1.2 to enhance per-iteration distractor mask prediction accuracy and training stability scene-agnostically in generalizable setting, as illustrated in Fig. 2.

### 4.1.1 REFERENCE-BASED MASK PREDICTION

This Mask Prediction module aims to enhance the $\mathcal{M}_{Rob}$ accuracy within each iterative various scene, ensuring that optimization efforts remain focused on more non-distractor areas. It is essential for generalization training, since Eq. 4 inevitably misclassifies certain target regions as distractor, particularly those presenting challenges for feed-forward inference, as in Fig. 3 6, which impedes the model's comprehension of geometric 3D consistency in Fig. 5. Our inspiration stems from an intuitive observation: *the 3DGS inferred from references maintains stable and accurate re-rendering results in non-distractor regions under reference views*. Specifically, we introduce a mask **filter** that harnesses non-distractor regions from re-rendered references (i.e. from references to infer 3DGS and render back to reference views) to identify and remove falsely labeled distractor regions in $\mathcal{M}_{Rob}$ under query view, based on the multi-view consistency of non-distractor static objects. Given $i \in N$ where $i$ denotes references and $\rho_{Ref}$ is a hyperparameter (set to 0.001), discussed in Fig. 10, re-rendered reference non-distractor masks $\mathcal{M}_{Ref_i}$ and corresponding projected query masks $\mathcal{M}_{Qry_i}$ are

$$\mathcal{M}_{Ref_i} = \mathbb{1}\left\{ \|\mathbf{I}_i - \mathcal{G}(\mathbf{P}_i)\|_2^2 < \rho_{Ref} \right\}, \quad \mathcal{M}_{Qry_i} = \mathcal{W}_{i \to T}(\mathcal{M}_{Ref_i}, \mathbf{D}_i, \mathbf{P}_i, \mathbf{P}_T, \mathbf{U}), \quad (5)$$

where $\mathbf{U}$ represents the camera intrinsic matrix of image pairs, $\mathbf{D}_i$ corresponds to the depth maps rendered from $\mathbf{P}_i$ utilizing a modified rasterization library, $\mathcal{W}_{i \to T}$ defines the image warping that projects each $\mathcal{M}_{Ref_i}$ from $\mathbf{P}_i$ to $\mathbf{P}_T$ using $\mathbf{D}_i$ and $\mathbf{U}$.

However, given the inherent noise presence in $\mathcal{M}_{Ref_i}$, $\mathcal{M}_{Qry_i}$ exhibits limited precision. To solve this problem, we incorporate a pre-trained segmentation model for mask filling, while designing a multi-mask fusion strategy to counteract noise-induced deviations. Following (Chen et al., 2024a; Otonari et al., 2024), we incorporate a state-of-the-art Entity Segmentation Model (Qi et al., 2022) to refine $\mathcal{M}_{Ref_i}$ into $\mathcal{M}_{Ref_i}^{En}$, which has the capability to fill different entities when the predicted distractor-region exceeds the threshold of total pixel count for that entity. After substituting $\mathcal{M}_{Ref_i}$ with $\mathcal{M}_{Ref_i}^{En}$ in Eq. 5, we use an intersection operation to fuse all ($N$) $\mathcal{M}_{Qry_i}$, then filter $\mathcal{M}_{Rob}$ based on it, $\mathcal{M}_Q = \left\{ \bigcap_{i=1}^{N} \mathcal{M}_{Qry_i} \right\} \bigcup \mathcal{M}_{Rob}$, obtaining the reference-based mask $\mathcal{M}_Q$, which can filter out misclassified regions in $\mathcal{M}_{Rob}$ to some extent, as shown in Fig. 3.

Here, we employ the intersection as a conservative strategy to ensure that the filtered-out regions in $\mathcal{M}_{Rob}$ are non-distractor regions acknowledged by all references and preserve the potential distractor are excluded from optimization regions, which is crucial for the training process. However, $\mathcal{M}_Q$ still exhibits limitations in accurate distractor masking, due to incorrect $\mathbf{D}_i$ prediction and view disparities, as illustrated in Fig. 3. Consequently, the distractor masks undergo further refinement in Sec. 4.1.2.

### 4.1.2 MASK REFINEMENT

Given $\mathcal{M}_Q$, a straightforward approach is to utilize a pre-trained segmentation model to refine noise regions and fill imprecise warping areas, as discussed with respect to $\mathcal{M}_{Ref_i}^{En}$. In contrast to references, $\mathcal{M}_Q$ contains distractor regions and disparity-induced errors arising from reference-query view variations, simultaneously, the latter being present in the query view but absent in all references and primarily occurring at the query image margins. Thus, before introducing the segmentation model, regions decoupling is essential, where the prediction of the disparity-induced error mask can follow a deterministic approach. Given $N$ one-filled masks $\mathcal{M}_i^{\mathbf{1}}$ corresponding to different reference

Figure 4: **Distractor-free Generalizable Inference Framework.** DGGS initially samples adjacent references from the scene-images pool and leverages trained DGGS for coarse 3DGS. Based on the **Reference Scoring mechanism**, masks and quality scores are computed for all pool images. These masks and scores subsequently guide reference selection and **Distractor Pruning** for fine 3DGS.

views $\mathbf{P}_i$, we warp them to the target view $\mathbf{P}_T$ as in Eq. 5. Then, the warped masks are merged using a union operation to ensure that these regions are absent from all references as Fig. 3.

$$\mathcal{M}_D = \bigcup_{i=1}^{N} \left\{ \mathcal{W}_{i \to T} \left( \mathcal{M}_i^1, \mathbf{D}_i, \mathbf{P}_i, \mathbf{P}_T, \mathbf{U} \right) \right\}. \tag{6}$$

Finally, we decouple $\mathcal{M}_D$ from $\mathcal{M}_Q$ and recombine them after introducing the segmentation model (Qi et al., 2022) and refining the distractor mask. The final refined mask, termed $\mathcal{M}$ in Fig. 3, substitutes $\mathcal{M}_{Rob}$ in Eq. 3 to mitigate distractor effects during training. Furthermore, we observe that if only Mask Refinement is used, without leveraging reference and our observation, the misclassification remains severe in Fig. 3, which verifies the importance of reference filter.

Additionally, in contrast to traditional distractor-free frameworks, references enable auxiliary supervision in the query view, providing guidance for occluded areas. Specifically, we re-warp $\mathcal{M}$ to reference views and utilize $\mathcal{M}_{Ref_i}^{En}$ to determine whether occlusion information is included. Therefore, our auxiliary loss $\mathcal{L}_A$ can focus on areas occluded from query view but visible from references.

$$\mathcal{L}_A = \sum_{N}^{i=1} \mathcal{W}_{T \to i} (1 - \mathcal{M}) \odot \mathcal{M}_{Ref_i}^{En} \odot \|\mathbf{I}_i - \mathcal{G}(\mathbf{P}_i)\|_2^2. \tag{7}$$

All pre-trained segmentation are pre-computed and cached. The final form of Eq. 3 is modified to:

$$\arg \min_{\boldsymbol{\theta}} \mathcal{M} \odot \|\mathbf{I}_T - \mathcal{G}(\mathbf{P}_T)\|_2^2 + \mathcal{L}_A. \tag{8}$$

## 4.2 DISTRACTOR-FREE GENERALIZABLE INFERENCE

Despite improvements in training robustness and mask prediction, DGGS's inference faces two key limitations: (1) insufficient references compromise reliable reconstruction of occluded and unseen regions; (2) persistent distractor in references inevitably appear as artifacts in synthesized novel views for feed-forward inference paradigm (encoder-decoder) in Eq. 2. To address these challenges, we propose a two-stage **Distractor-free Generalizable Inference** framework in Fig. 4. The first stage employs a **Reference Scoring** mechanism to score candidate references from the images pool, facilitating the selected references with minimal distractor and disparity. The second stage uses a **Distractor Pruning** module to suppress the remaining artifacts.

**Reference Scoring Mechanism.** The objective of the first inference stage is to select references with minimal distractor and disparity among the predefined scene-images pool, adjacent $K$ views ($K \geq N$) sampled in the test scenes. Therefore, we propose a Reference Scoring mechanism based on the pre-trained DGGS in Sec. 4.1. Specifically, it first samples $N$ adjacent references from the scene-images pool $\{\mathbf{I}\}_P^K$ for coarse 3DGS prediction. We then designate unselected views from $\{\mathbf{I}\}_P^K$ as query for mask prediction $\{\mathcal{M}\}^{K-N}$, while the distractor masks of the chosen reference views are $\{\mathcal{M}_{Ref}^{En}\}^N$. All image masks in pool are scored by Eq. 9, in which top $N$ images are selected in next stage,

$$\{\mathbf{I}_i\}^N = \{\mathbf{I}_i\}_P^K \mid i \in \arg \max_{N} \left\{ \mathcal{S} \left( \{\mathcal{M}\}^{K-N}; \{\mathcal{M}_{Ref}^{En}\}^N \right) \right\}. \tag{9}$$

where $\mathcal{S}$ is the pixel-wise summation for each mask. In practice, besides distractor size, the extrinsics of candidate images are also crucial reference scoring factors due to disparity. Thanks to the discussion of disparity-induced error masks in Sec. 4.1.2, we can directly utilize the count of positive pixels in $\mathcal{M}$ as the single criterion, which selects references that provide better coverage of query view, as shown

Table 1: **Quantitative Experiments for distractor-free Generalizable 3DGS** under RobustNeRF. * denotes pre-trained models, + indicates baseline models augmented with existing masking methods.

| Methods | Statue (RobustNeRF) | | | Android (RobustNeRF) | | | Mean (Five Scenes) | | | Train Data |
|---|---|---|---|---|---|---|---|---|---|---|
| | PSNR↑ | SSIM↑ | LPIPS↓ | PSNR↑ | SSIM↑ | LPIPS↓ | PSNR↑ | SSIM↑ | LPIPS↓ | |
| Pixelsplat* (2024 CVPR) | 18.65 | 0.673 | 0.254 | 17.98 | 0.557 | 0.364 | 20.10 | 0.704 | 0.279 | Pre-Train |
| Mvsplat* (2024 ECCV) | 18.88 | 0.670 | **0.225** | 18.24 | 0.586 | 0.301 | 20.03 | 0.722 | 0.255 | on Re10K |
| Pixelsplat (2024 CVPR) | 15.49 | 0.378 | 0.531 | 16.34 | 0.331 | 0.492 | 16.02 | 0.422 | 0.511 | |
| Mvsplat (2024 ECCV) | 15.05 | 0.412 | 0.391 | 16.17 | 0.509 | 0.381 | 15.45 | 0.515 | 0.426 | |
| +RobustNeRF (2023 CVPR) | 16.17 | 0.463 | 0.382 | 16.46 | 0.470 | 0.411 | 17.11 | 0.534 | 0.400 | Re-Train on |
| +On-the-go (2024 CVPR) | 14.73 | 0.366 | 0.522 | 15.05 | 0.440 | 0.472 | 15.44 | 0.476 | 0.526 | Distractor- |
| +NeRF-HuGS (2024 CVPR) | 18.21 | 0.694 | 0.266 | 18.33 | 0.640 | 0.299 | 19.18 | 0.700 | 0.283 | Datasets |
| +HybridGS (CVPR 2025) | 17.16 | 0.540 | 0.369 | 16.37 | 0.517 | 0.375 | 17.82 | 0.556 | 0.388 | |
| +SLS (Arxiv 2024) | 18.11 | 0.695 | 0.270 | 18.84 | 0.662 | 0.282 | 19.29 | 0.709 | 0.286 | |
| DGGS-TR (w/o Inference Part) | 19.68 | 0.700 | 0.238 | 19.58 | 0.653 | 0.286 | 21.02 | 0.738 | 0.242 | |
| DGGS (Our) | **20.78** | **0.710** | 0.233 | **20.93** | **0.711** | **0.236** | **21.74** | **0.758** | **0.237** | |

Table 2: **Ablation** for DGGS-TR and DGGS.

| Methods | Mean (Five Scenes) | | |
|---|---|---|---|
| | PSNR↑ | SSIM↑ | LPIPS↓ |
| Baseline (Mvsplat) | 15.45 | 0.515 | 0.426 |
| +Robust Mask | 17.11 | 0.534 | 0.400 |
| ++Ref-based Mask Prediction | 20.35 | 0.701 | 0.283 |
| +++Mask Refinement (DGGS-TR) | **21.02** | **0.738** | **0.242** |
| w/o Reference Entity Segmantation | 20.79 | 0.733 | 0.248 |
| w/o Aux Loss | 20.64 | 0.725 | 0.253 |
| DGGS-TR | 21.02 | 0.738 | 0.242 |
| + Reference Scoring mechanism | 21.47 | 0.749 | 0.242 |
| ++ Distractor Pruning (DGGS) | **21.74** | **0.758** | **0.237** |

Table 3: **Comparison** of Fine-Tuned models.

| Methods | Arcdetriomphe | | | Mountain | | |
|---|---|---|---|---|---|---|
| | PSNR↑ | SSIM↑ | LPIPS↓ | PSNR↑ | SSIM↑ | LPIPS↓ |
| Mvsplat*+FT | 23.58 | 0.841 | 0.08 | 17.06 | 0.622 | 0.202 |
| Mvsplat*+SLS-FT | 27.19 | 0.916 | 0.084 | 22.03 | 0.698 | 0.160 |
| Mvsplat*+DGGS-FT | 28.61 | 0.922 | 0.068 | 23.00 | 0.723 | 0.144 |
| (Mvsplat+SLS)*+SLS-FT | 24.18 | 0.887 | 0.095 | 20.77 | 0.645 | 0.189 |
| SLS(Single Scene Train) | - | - | - | 22.53 | 0.77 | 0.18 |
| DGGS-TR*+DGGS-FT | **29.04** | **0.931** | **0.058** | **23.85** | **0.787** | **0.128** |

Table 4: **Comparison** on Efficiency.

| Methods | Pixelsplat | Mvsplat | DGGS (Two Stage Inference) |
|---|---|---|---|
| Rendering Time (s) | 0.160 | **0.084** | 0.148 [w pre-segmentation 0.111] |

in Fig. 7 8. In the second stage, we employ top-ranked images as references for fine reconstruction, effectively reselecting the candidate images in the pool without increasing GPU memory. Although this approach successfully handles distractor-heavy references, it comes at the cost of decreased rendering efficiency. Optionally, we mitigate this by halving the image resolution in the first stage.

**Distractor Pruning.** Although cleaner references are selected, obtaining $N$ distractor-free images in the wild is virtually impossible. These residual distractor propagate via the gaussian encoding-decoding process in Eq. 2, manifesting as phantom splats in rendered query view, as shown in Fig. 8. Therefore, we propose a Distractor Pruning protocol in the second inference stage, which is readily implementable given the reference distractor masks. Instead of direct masking on the references, which affects one-to-one mapping between pixels and gaussian primitives in Eq. 2, we selectively prune gaussian primitives within the 3D space by directly removing decoded attributes in distractor regions while preserving the remaining components. However, when references exhibit a large amount of commonly occluded regions, the pruning strategy induces white speckle artifacts. Consequently, based on projected masks, DGGS implements the pruning strategy only in scenarios where it is not considered a common occluded region for all references, which is discussed in Sec. 6.

## 5 EXPERIMENTS

### 5.1 SYNTHETIC DATASET

To train and evaluate DGGS, beyond the real-world datasets On-the-go (Ren et al., 2024) and RobustNeRF Dataset (Sabour et al., 2023), we construct a synthetic dataset to augment the number of distractor scenes based on Re-10K and ACID, with details provided in AppendixA.

### 5.2 COMPARATIVE EXPERIMENTS

**Benchmark:** Our Distractor-free Generalizable training and inference paradigms can be seamlessly integrated with existing generalizable 3DGS frameworks. We adopt Mvsplat (Chen et al., 2024b) as our baseline model. Extensive comparisons are conducted against existing works trained under same settings and distractor datasets, including: **(1) retraining** existing generalization methods (Chen et al., 2024b; Charatan et al., 2024), and **(2) retraining** Mvsplat (Chen et al., 2024b) '+' different mask prediction strategies from distractor-free approaches (Ren et al., 2024; Sabour et al., 2023; Chen et al., 2024a; Sabour et al., 2024). Although all current distractor-free works are scene-specific, most of them focus on applying distractor masks in loss function (as discussed in Sec. 2.2), so we can directly transfer them to the query rendering loss under our baseline. Specifically, during the training

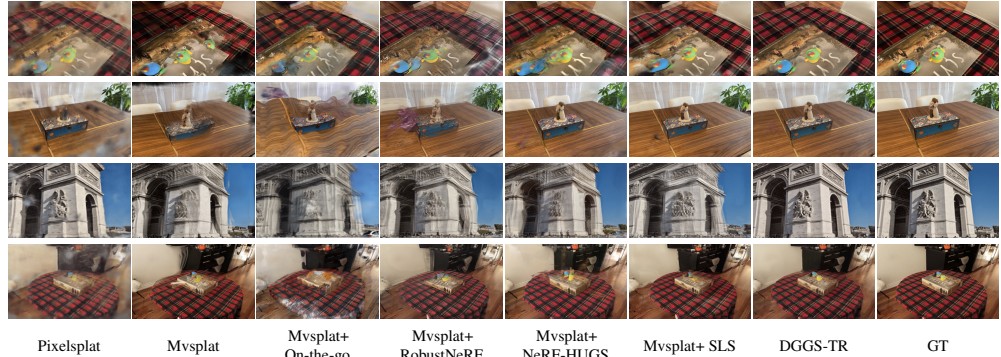

| Pixelsplat | Mvsplat | Mvsplat+ On-the-go | Mvsplat+ RobustNeRF | Mvsplat+ NeRF-HUGS | Mvsplat+ SLS | DGGS-TR | GT |

Figure 5: **Qualitative Comparison of Re-trained Existing Methods** across unseen scenes.

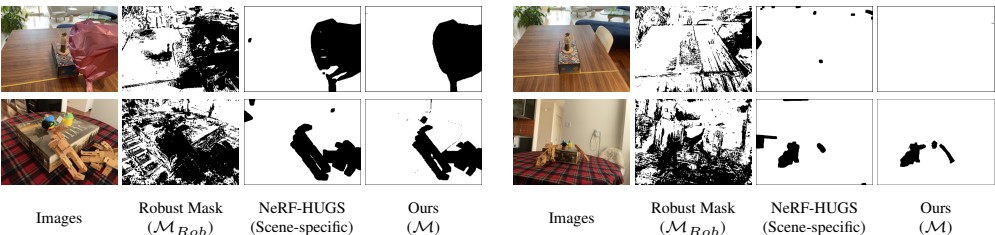

| Images | Robust Mask ($\mathcal{M}_{Rob}$) | NeRF-HUGS (Scene-specific) | Ours ($\mathcal{M}$) | Images | Robust Mask ($\mathcal{M}_{Rob}$) | NeRF-HUGS (Scene-specific) | Ours ($\mathcal{M}$) |

Figure 6: **Qualitative Comparison** for our masks prediction vs. scene-specific results.

phase, we combine these mask prediction methods from existing works and replace $\mathcal{M}_{Rob}$ in Eq .3. Furthermore, **(3)** we compare with existing **pre-trained** generalizable models (distractor-free).

**Quantitative and Qualitative Experiments:** Tab. 1, Fig. 5 and Fig. 7 quantitatively and qualitatively compares DGGS-TR (only TRaining in Sec. 4.1 without two stages inference in Sec. 4.2) and DGGS with existing methods. The results are analyzed from three aspects: re-training models, pre-training models and models after fine-tuning.

**For Re-train Model**: Tab. 1 and Fig. 5 demonstrate that our training paradigm significantly enhances training robustness, outperforming existing generalizable 3DGS algorithms, **PSNR: 21.02 vs. 15.45**, under same training conditions. Compared to introducing existing scene-specific distractor-free approaches, where overaggressive distractor prediction degrades reconstruction fidelity, DGGS-TR exhibits enhanced reconstruction detail and 3D consistency, **PSNR: 21.02 vs. 19.29**, through precise distractor prediction, which is further validated in Fig. 6. Similarly, the qualitative results in Fig. 5 show training instability and significant performance drops, even without distractors in some test references. It confirms that distractor presence severely impairs model learning of geometric 3D consistency, aligning with our motivation. Additionally, experiments also show that DGGS possesses cross-scene feed-forward inference capabilities that existing distractor-free methods lack.

**For Pre-train Model:** Tab. 1 reports comparisons between DGGS-TR, DGGS and existing pre-train generalizable models. Compared with pre-train models on distractor-free data, DGGS-TR exhibits superior performance even with training on limited distractor scenes, mainly due to mitigating the inference scene *distractor impacts* and *domain gap*. Fig. 7 also illustrates similar findings: DGGS-TR effectively attenuates partial distractor effects in the 3D inconsistency regions. Furthermore, after introducing our inference paradigm, DGGS demonstrates 'pseudo-completion' and artifact removal capabilities, which benefit from the **Reference Selection** mechanism that can choose references with less distractor as well as disparity and the **Distractor Pruning** during inference process as in Fig. 8. To further validate effectiveness, we compare DGGS's results on synthetic data, which maintains the same data domain as the pre-trained MVSplat* except for inserted distractor. In Fig. 9, DGGS shows better resistance to distractor, effectively mitigating artifact. More results are shown in Appendix.

**After Fine-tuning:** To further demonstrate DGGS's capability, we conduct fine-tuning experiments on the different pre-trained models (including Mvsplat*, (Mvsplat+SLS)* and DGGS-TR* corresponding Line 2, 8, and 9 in Tab. 1) and training strategies (including FT, SLS-FT and DGGS-FT corresponding Mvsplat (Chen et al., 2024b), SLS (Sabour et al., 2024) and Ours). We fine-tune using the 'clutter' data and evaluate on the 'extra' data in test scenes. For fairness in comparison, our inference framework is not employed. In Tab. 3, DGGS-TR demonstrates significant improvements

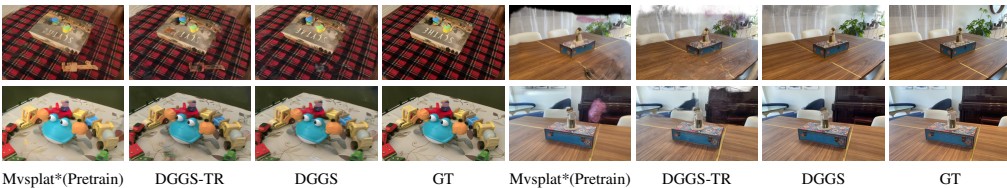

Mvsplat*(Pretrain)     DGGS-TR     DGGS     GT     Mvsplat*(Pretrain)     DGGS-TR     DGGS     GT

Figure 7: **Qualitative Comparison of Pre-trained Models**, DGGS-TR and DGGS.

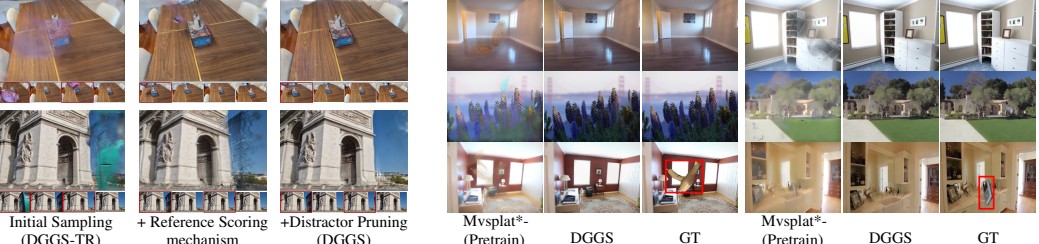

Initial Sampling (DGGS-TR)    + Reference Scoring mechanism    +Distractor Pruning (DGGS)      Mvsplat*-(Pretrain)    DGGS    GT    Mvsplat*-(Pretrain)    DGGS    GT

Figure 8: **Ablation for Inference Strategy**.      Figure 9: **Comparison** on synthetic data.

over existing models and strategies, attributed to our reliable distractor prediction. Furthermore, we find that DGGS's single-scene fine-tuning performance is even better than SLS's single-scene training (metrics reported by SLS (Sabour et al., 2024)). This benefits from DGGS's pretraining on large-scale datasets and its reference-based strategy for distractor pridiction.

**Efficiency:** Tab. 4 shows the efficiency comparison. Due to the segmentation model and two-stage inference, DGGS has a slight decrease in efficiency compared to (Chen et al., 2024b), but mitigates distractor and obtains masks. It can be boosted by reducing segmentation resolution.

**Re-rendered Reference Accuracy:** One of the core insights of DGGS is to leverage the more stable reference re-render as an additional filter to assist query distractor mask prediction. Most existing works exhibit this trend, such as Mvsplat* (Chen et al., 2024b). To verify the stability of re-rendering during the generalizable 3DGS entire training process, we compare the performance of query and re-rendering reference views in Tab. 6. From the experimental results on DGGS, it can be easily observed that from the beginning of DGGS training, reference re-rendering exhibits more stable performance relative to the query view. This characteristic allows reference re-rendering to effectively guide distractor mask prediction for the query view. Furthermore, we simultaneously observe that as the training process progresses, the performance gap between the two gradually narrows.

## 5.3 ABLATION STUDIES

### 5.3.1 ABLATION ON TRAINING FRAMEWORK

The upper section of Tab. 2 and Fig. 3 show the effectiveness of each component in DGGS training paradigm. The Reference-based Mask Prediction combined with Mask Refinement successfully mitigates the tendency of over-predicting targets as distractor that occurred in the original $\mathcal{M}_{Rob}$. Within the Mask Refinement module, Auxiliary Loss demonstrates remarkable performance, while Mask Decoupling and Entity Segmentation contribute substantial improvements to the overall framework. Furthermore, our analysis in Fig. 6 reveals that DGGS-TR achieves scene-agnostic mask prediction capabilities, with feed-forward inference results superior to single-scene trained models (Chen et al., 2024a). Note that, for fair evaluation, all masks are predicted under DGGS-TR based same references, without the involvement of our inference framework.

### 5.3.2 ABLATION ON INFERENCE FRAMEWORK

The bottom portion of Tab. 2, Fig. 7, and Fig. 8 analyze the component effectiveness within the inference paradigm. Results indicate that although the Reference Scoring mechanism alleviates the impact of distractor in references by re-selection, red box represent distractor references in Fig. 8, certain artifacts remain unavoidable. Then, our Distractor Pruning strategy effectively mitigates these residual artifacts. We also analyze how the choice of $K$ and $N$ in DGGS, the sizes of scene images pool and references, affects inference results in Fig. 10. Generally, larger values of $K$ yield better performance up to $2N$, beyond which performance plateaus, likely due to increased significant view disparity in the images pool. A similar situation occurs with $N$, where excessive references actually

Table 5: **Ablation Study on Segmentation Module** under RobustNeRF dataset.

| Method | PSNR↑ | SSIM↑ | LPIPS↓ |
|---|---|---|---|
| DGGS-TR (w/o Segmentation) | 20.67 | 0.730 | 0.251 |
| DGGS-TR (SAM2) | 21.07 | 0.740 | 0.240 |
| DGGS-TR (w/ Segmentation) | 21.02 | 0.738 | 0.242 |
| DGGS (w/o Segmentation) | 21.41 | 0.749 | 0.240 |
| DGGS (SAM2) | **21.77** | 0.758 | **0.236** |
| DGGS (w/ Segmentation) | 21.74 | **0.758** | 0.237 |

Table 6: **Query v.s. Re-rendering Reference Analysis** comparing PSNR at different training iterations.

| Method | Train Stage | Query | Ref Re-render |
|---|---|---|---|
| Trained Mvsplat* | Done | 26.02 | 29.71 |
| DGGS-TR | 200 | 12.13 | 17.82 |
| DGGS-TR | 1000 | 16.36 | 20.38 |
| DGGS-TR | 10000 | 22.11 | 25.65 |
| Trained DGGS-TR | Done | 26.51 | 28.02 |

increase distractor and disparity. And, Tab. 13 explores why the Reference Scoring mechanism brings performance improvements. A potential possibility is our ability to access more references during inference (Chen et al., 2024b). Therefore, we directly select more references in DGGS-TR. Experimental findings indicate, incorporating additional references directly will introduce novel artifacts. It proves that the quality of references takes precedence over quantity in our setting, and the proposed Reference Scoring mechanism can filter out higher-quality references.

### 5.3.3 ABLATION ON PRE-TRAINED SEGMENTATION MODEL

Pre-trained entity segmentation has been widely used in previous scene-specific distractor-free studies(Catley-Chandar et al., 2024; Chen et al., 2024a). We further analyze the role of the pretrained segmentation model in the training and inference stages of DGGS in Tab. 5 and Tab. 2. It is not difficult to find that whether during training or inference, the pretrained segmentation model provides limited performance improvement compared to other modules, such as reference-based mask prediction. This is attributed to the fact that the core factors for stable and generalizable 3DGS training are 3D consistency among reference views, rather than fine-grained distractor segmentation boundaries. During the inference process, multi-reference inference and the reference scoring mechanism similarly contribute to enhancing the robustness of DGGS with respect to segmentation results. Furthermore, we also replace the Entity Segmentation model with the more robust segmentation model SAM2 (Ravi et al., 2024). Although it is capable of outputting more accurate distractor segmentation masks, experiments demonstrate that its impact on overall reconstruction performance is limited.This also further demonstrates DGGS's robustness to predefined segmentation results in terms of reconstruction performance. More discussion about the segmentation model is presented in the Appendix. B.4

## 6 CONCLUSION

Distractor-free Generalizable 3D Gaussian Splatting presents a practical challenge, offering the potential to mitigate the limitations imposed by distractor scenes on generalizable 3DGS while addressing the scene-specific training constraints of existing distractor-free methods. We propose novel training and inference paradigms that alleviate both training instability and inference artifacts from distractor data. Extensive experiments across diverse scenes validate our method's effectiveness and demonstrate the potential of the reference-based paradigm in handling distractor scenes. We envision this work laying the foundation for future community discussions on Distractor-free Generalizable 3DGS and potentially extending to address real-world 3D data challenges in broader applications.

**Limitation:** DGGS mitigates distractor but experiences performance degradation under extensive common occlusions. Since DGGS does not have additional generative completion capabilities, it will exhibit phenomena such as speckles when facing situations with common occlusions among all references, as shown in Fig. 16. Generative models offer a potential solution. In addition, the sacrifice in efficiency is unavoidable, which can be mitigated through lighter segmentation models and resolution.

## REPRODICIBILITY STATEMENT

We have made every effort to ensure our work is reproducible. Our experiments are conducted on the public Mvsplat benchmarks. The methodology for constructing the DGGS is detailed in Sec.4. To further facilitate replication, we provide comprehensive training configurations and implementation details in appendix A. The source code for our framework and experiments will be made publicly available upon publication.

ACKNOWLEFGEMENTS

This work is supported in part by the National Natural Science Foundation of China (62276128, 62192783), Jiangsu Natural Science Foundation(BK20243051), Jiangsu Science and Technology Major Project (BG2024031), the Fundamental Research Funds for the Central Universities(14380128, KG202514), a grant from the NSFC/RGC Collaborative Research Scheme sponsored by the Research Grants Council of the Hong Kong Special Administrative Region, China and National Natural Science Foundation of China (Project No. CRS_ HKUST605/25), and the Collaborative Innovation Center of Novel Software Technology and Industrialization.

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

# A EXPERIMENTAL DETAILS

## A.1 REAL AND SYNTHETIC DATASET

In accordance with existing generalization frameworks, DGGS is trained in a cross-scene setting with distractor and evaluated in novel unseen distractor scenes to simulate real-world scenarios. Specifically, we utilize two widely used mobile-captured datasets: On-the-go (Ren et al., 2024) and RobustNeRF (Sabour et al., 2023), containing multiple distractor-scenes in outdoor and indoor environments, respectively. For fair comparison in real data, we train all models on On-the-go outdoor-scenes except *Arcdetriomphe* and *Mountain*, which, along with the RobustNeRF indoor-scenes, serve as test scenes. In addition to real data, we also construct synthetic datasets for additional verification.

### A.1.1 REAL DATASET

DGGS primarily addresses distractor challenges in both training and inference under generalization settings. To validate the reliability of DGGS in the wild, we conduct experiments on existing real-world distractor datasets, which combine the On-the-go (Ren et al., 2024) (outdoor scenes) and RobustNeRF (Sabour et al., 2023) (indoor scenes) datasets. These datasets are captured by mobile devices and constitute the most widely utilized benchmark for scene-specific distractor-free methods, as discussed in Sec. 2.2.

**On-the-go Dataset:** It, captured using an iPhone 12, Samsung Galaxy S22, and DJI Mini 3 Pro drone, provides a diverse and dynamic range of distractor with varying occlusion ratios (5% to 30%). The dataset consists of 12 casually captured sequences, including Drone, Patio, and ArcdeTriomphe, most of which are outdoor scenes.

**RobustNeRF Dataset:** It comprises four natural indoor scenes —two in an apartment and two in a robotics lab—captured with controlled settings and varying complexities, including dynamic distractor and fixed camera parameters.

We downsample all images' resolution to $192 \times 256$ in all experiments. As described in Sec. 5.1, we designate all scenes from On-the-go dataset except *Arcdetriomphe* and *Mountain* as the training set, while these two scenes along with the RobustNeRF dataset serve as evaluation scenes. For evaluating the robustness against distractor, we utilize the 'extra' views from the On-the-go dataset and the 'clean' views from the RobustNeRF dataset (except for the *Crab1*) as inference views, and compute corresponding evaluation metrics to assess performance. Note that all reference-query pair selection strategies ensure distinct poses between them, particularly for scenes with clean-clutter paired configurations.

### A.1.2 SYNTHETIC DATASET

To further validate our novel task: Distractor-free Generalizable 3D Gaussian Splatting, we construct numerous synthetic scenes based on Re10K (Zhou et al., 2018) and ACID (Liu et al., 2021) datasets that are widely used in the field of generalizable reconstruction (Chen et al., 2024b). These datasets, sourced from YouTube, contain numerous multi-view images of static indoor and outdoor scenes with corresponding camera poses that we use as the foundation to construct distractor on them. For distractor, we select the COCO (Lin et al., 2014) dataset as our source, as it contains numerous object categories with their corresponding semantics and masks.

Specifically, we build a distractor library using COCO masks and randomly introduce different distractor to random positions with random rotations in the multi-view images of various static scenes. It's worth noting that for multi-view images of a same scene, we ensure the inserted distractor is the same object, which better aligns with real-world scenarios. Unlike real-world scene data, we can control the frequency of distractor appearances and their coverage area in images, which provides further validation for DGGS. We ensure that 70% of the images contain inserted distractors while the remaining images remain distractor-free. Additionally, through scaling the distractors, we control their occupied area to consistently range between 0% and 30% of the entire scene. During evaluation, we maintain consistent settings with the Real Dataset, including the selection of references and queries and the evaluation methodology.

Figure 10: **Performance Comparison under Different Inputs, and Ablation Studies for $\rho_{Ref}$.**

## A.2 TRAINING AND EVALUATION SETTING

In all experiments, we set the number of references $N$=4 and the size of scene-images pool $K$=8, which are discussed in Fig. 10. During all training, query views and reference views are selected, regardless of 'clutter' or 'extra'. In evaluation phase, we utilize all 'extra' images and 'clear' images with a stride of eight as query views for On-the-go (*Arcdetriomphe* and *Mountain*) and all RobustNeRF scenes. For references in evaluation, we construct the scene-images pool using views closest to the query view, ensuring inclusion of both distractor-containing and few distractor-free data to validate the effectiveness of Reference Scoring mechanism. Note that this setting is only for validation and evaluation purposes. In practical applications, the scene-images pool can be constructed using any adjacent views, independent of the query view and distractor presence. Finally, we compute the average PSNR, SSIM, and LPIPS metrics across all scenes on all query renders. More details are described in Appendix.

### A.2.1 EXPERIMENTAL SETTING

To ensure a fair comparison, we maintain hyperparameters, experimental settings, and evaluation method across all re-training experiments. All re-trained models are implemented with PyTorch 2.1.0, and all experiments are conducted using Nvidia RTX 3090 GPUs with CUDA 12.1 and trained for 10K iterations.

### A.2.2 OTHER RE-TRAINED COMPARATIVE METHODS

In addition to maintaining consistent experimental settings as mentioned above, we conduct comparative experiments by integrating distractor-free approaches with generalizable 3DGS methods. Specifically, we incorporate existing distractor-free mask prediction and loss function improvements into our baseline model, Mvsplat (Chen et al., 2024b). This integration approach is viable primarily because existing distractor-free methods largely emphasize residual-based mask prediction, facilitating seamless incorporation into the query residual loss framework within generalizable configurations. The integration methodology varies according to different approaches, as detailed below:

**RobustNeRF (Sabour et al., 2023) and SLS (Sabour et al., 2024):** We directly incorporate RobustNeRF's and SLS's mask prediction component to guide the MSE loss in Mvsplat (Chen et al., 2024b).

**NeRF-On-the-go (Ren et al., 2024):** While maintaining all other components unchanged, we similarly incorporate DINOv2 (Oquab et al., 2023) to predict uncertainty maps for guiding the MSE loss in Mvsplat (Chen et al., 2024b).

**NeRF-HuGS (Chen et al., 2024a):** To maintain maximum consistency with the original method, we incorporate both SAM (Kirillov et al., 2023) and predefined Colmap (Schonberger & Frahm, 2016) to guide mask prediction in MvSplat (Chen et al., 2024b)'s MSE loss. We exclude Nerfacto (Tancik et al., 2023) from our implementation, as the per-scene pretraining becomes impractical under our iterative scene transition setting.

## B ADDITIONAL RESULTS AND ANALYSIS

This section presents extensive experimental results, both qualitative and quantitative, across various scenarios, with further analysis validating the efficacy of DGGS.

Table 7: **Quantitative Experiments** for DGGS under single-scene setting

| Methods | Statue (RobustNeRF) | | |
| --- | --- | --- | --- |
| | PSNR↑ | SSIM↑ | LPIPS↓ |
| NeRF-W (Martin-Brualla et al., 2021) (CVPR 2021) | 18.91 | 0.616 | 0.369 |
| HA-NeRF (Chen et al., 2022) (CVPR 2022) | 18.67 | 0.616 | 0.367 |
| RobustNeRF (Sabour et al., 2023) (CVPR 2023) | 20.60 | 0.758 | 0.154 |
| On-the-go (Ren et al., 2024) (CVPR 2024) | 21.58 | 0.77 | 0.24 |
| NeRF-HuGS (Chen et al., 2024a) (2024 CVPR) | 21.00 | 0.774 | 0.135 |
| SLS (Sabour et al., 2024) (Arxiv 2024) | 22.69 | 0.85 | 0.12 |
| DGGS-TR | 21.57 | 0.769 | 0.187 |
| DGGS-TR (+ Distractor-free References) | **30.50** | **0.952** | **0.052** |

Table 8: **Memory Usage and Time** under different configurations.

| $N$ ($K$) | Resolution | Memory | Time (w/o Seg) |
| --- | --- | --- | --- |
| 4 (8) | 252 | 10G | 0.111s |
| 4 (8) | 504 | 16G | 0.277s |
| 6 (12) | 252 | 12G | 0.203s |

Table 9: **Ablation Study on Segmentation Model for Mask Prediction.**

| Method | PSNR↑ | SSIM↑ | LPIPS↓ |
| --- | --- | --- | --- |
| DGGS-TR (Feature Consistency) | 20.85 | 0.733 | 0.242 |
| DGGS-TR (Clustering) | 20.92 | 0.736 | 0.245 |
| DGGS-TR (Our) | **21.02** | **0.738** | **0.242** |

## B.1 QUANTITATIVE RESULTS IN MORE REAL-SCENES

Qualitative comparisons of DGGS in RobustNeRF scenes are presented in Tab. 1. Furthermore, we present results on scenes *Arcdetriomphe* and *Mountain* from the On-the-go dataset in Tab. 12. It is worth noting that we omit the discussion of DGGS (with Our Inference) for these scenes quantitatively as their 'extra' views used for evaluation still contain partial distractor. The qualitative comparisons for them are extensively discussed in subsequent sections. Extensive quantitative evaluations across diverse scenes validate the superiority of DGGS, notably outperforming existing generalizable methods that suffer from distractor-induced uncertainties. Although augmenting the baseline with scene-specific distractor-free methods improves stability in some scenes, DGGS's scene-agnostic mask prediction capability demonstrates superior performance in generalizable distractor-free 3DGS reconstruction.

## B.2 MEMORY USAGE AND TIME ANALYSIS

To further analyze the memory Usage and time for DGGS, we additionally report the efficiency under different $N(K)$ and different resolutions in Tab. 8. The impact of these factors on efficiency is similar to works like Mvsplat (Chen et al., 2024b); that is, as the resolution and $N$ increase, the inference time and memory usage of DGGS also increase accordingly.

## B.3 ABLATION STUDY FOR MASK FUSION

DGGS adopts a conservative intersection operation reference multi-mask fusion strategy in Sec.4.1.1, which effectively reduces the impact of distractors during generalizable 3DGS training. To further verify the effectiveness of this method, we additionally discuss some variant mask fusion approaches. In Tab.11, we modify the original intersection-based mask fusion strategy to require at least 50% of the references to agree, meaning that if two references indicate that the corresponding location is a static region, it can be considered static. The experimental results show that this leads to a decline in the training model's performance, which may be due to distractor regions being misclassified as static regions. This further demonstrates what we mention earlier: incorrectly classifying dynamic regions as static has a greater impact on generalizable 3DGS training than incorrectly classifying static regions as dynamic. Furthermore, we also discuss utilizing soft weighting as an alternative to the strict intersection of all references during inference. A straightforward approach is to employ the view angle as an additional condition for weighted fusion of all masks, whereby references that are more distant from the target view are assigned proportionally lower weights. Experiments

Table 10: **Quantitative Experiments** for DGGS on Synthetic Testing Scenes

| Methods | Synthetic Testing Scenes (Mean) | | |
| --- | --- | --- | --- |
| | PSNR↑ | SSIM↑ | LPIPS↓ |
| Mvsplat* (Pre-train on Re-10K) (Chen et al., 2024b) | 18.02 | 0.758 | 0.220 |
| DGGS (Real → Synthetic) | 26.51 | 0.912 | 0.105 |
| DGGS (Fine-tuning on Synthetic Training Scenes) | **29.66** | **0.949** | **0.059** |

Table 11: **Ablation Study for Mask Fusion.**

| Method | PSNR↑ | SSIM↑ | LPIPS↓ | Method | PSNR↑ | SSIM↑ | LPIPS↓ |
| --- | --- | --- | --- | --- | --- | --- | --- |
| DGGS-TR (50%) | 20.70 | 0.734 | 0.250 | DGGS (View Angles) | 21.60 | 0.752 | 0.240 |
| DGGS-TR (Our) | **21.02** | **0.738** | **0.242** | DGGS (Our) | **21.74** | **0.758** | **0.237** |

demonstrate that distractor masks are view-independent, and such fusion leads to additional distractor artifacts in some cases. This means that if a distractor appears in any reference view, regardless of its distance from the target view, it negatively affects the reconstruction quality of the entire scene.

## B.4 ABLATION STUDY FOR SEGMENTATION MODELS

To further discuss the importance of pretrained segmentation models, we additionally introduce feature consistency loss terms or Cclustering strategies to replace the segmentation model. These strategies are verified to have certain robustness against some misclassified regions for distractor. From the results in Tab. 9, it can be found that the pretrained segmentation model can be replaced during the training process without causing serious impact on reconstruction performance. This is similar to our previous analysis, that is, DGGS mainly relies on 3D consistency among references to determine the distractor and stabilize the training process, rather than accurate masks. However, the difference from these methods is that they can hardly output accurate distractor masks additionally, whereas DGGS is able to simultaneously achieve distractor mask prediction during the reconstruction.

## B.5 QUANTITATIVE RESULTS IN SYNTHETIC-SCENES

Corresponding to Fig.10 in the main text, we quantitatively discuss the performance comparison of DGGS in synthetic scenes. We discuss two training settings for DGGS, including training on real scenes and directly inferring on synthetic scenes, as well as fine-tuning the model trained on real scenes using synthetic scenes. To test on unseen scenes, the scenes used for fine-tuning are kept inconsistent with the scenes used for inference. Tab. 10 demonstrates the generalization ability of DGGS despite the significant domain gap between the real scenes used for training and the synthetic scenes used for testing. Meanwhile, this also shows that the pre-trained Mvsplat* struggles to handle the impact brought by distractors, which may be caused by 3D inconsistencies. We further verify this in subsequent qualitative experiments, as shown in Fig. 11.

## B.6 QUALITATIVE EXPERIMENTS

Fig. 14 and Fig. 15 present comprehensive qualitative comparisons of DGGS-TR and DGGS against existing methods. Through extensive visualization results, DGGS-TR demonstrates superior stability, particularly in high-frequency regions, compared to other methods trained on distractor-rich data. However, its performance deteriorates when processing reference images containing substantial distractor. While marginally outperforming pre-trained models (Chen et al., 2024b), it continues to face challenges with artifact suppression and hole elimination. Leveraging our inference strategy, DGGS effectively addresses these challenges. Extensive examples in Fig. 15 validate its ability to mitigate artifacts and holes, producing cleaner scene representations.

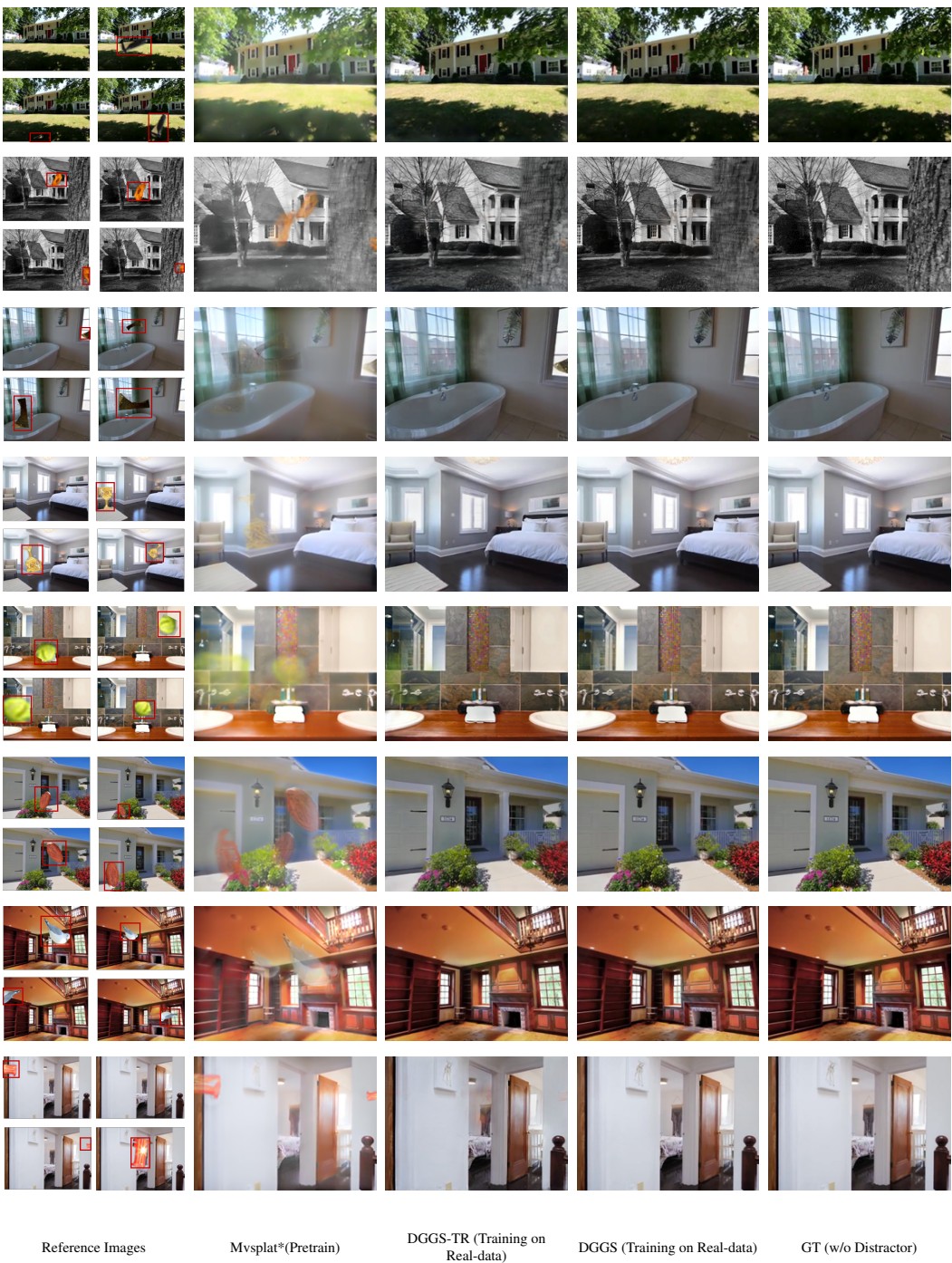

Reference Images    Mvsplat*(Pretrain)    DGGS-TR (Training on Real-data)    DGGS (Training on Real-data)    GT (w/o Distractor)

Figure 11: **Qualitative Comparison on Synthetic scenes** with Same References.

Table 12: **Quantitative Experiments** for distractor-free Generalizable 3DGS under *Arcdetriomphe* and *Mountain* scenes.

| Methods | Arcdetriomphe | | | Mountain | | |
|---|---|---|---|---|---|---|
| | PSNR↑ | SSIM↑ | LPIPS↓ | PSNR↑ | SSIM↑ | LPIPS↓ |
| Pixelsplat (Charatan et al., 2024) | 15.15 | 0.311 | 0.435 | 15.43 | 0.443 | 0.441 |
| Mvsplat (Chen et al., 2024b) | 14.96 | 0.341 | 0.401 | 13.73 | 0.253 | 0.419 |
| MVSgaussian (Liu et al., 2025) | 14.28 | 0.528 | 0.531 | - | - | - |
| +RobustNeRF (Sabour et al., 2023) | 15.98 | 0.390 | 0.372 | 14.61 | 0.276 | 0.445 |
| +On-the-go (Ren et al., 2024) | 14.67 | 0.354 | 0.432 | 14.23 | 0.287 | 0.436 |
| +NeRF-HuGS (Chen et al., 2024a) | 19.02 | 0.699 | 0.250 | 15.26 | 0.463 | 0.355 |
| +SLS (Sabour et al., 2024) | 19.28 | 0.716 | 0.227 | 15.14 | 0.443 | 0.388 |
| DGGS-TR | **20.32** | **0.737** | **0.214** | **16.37** | **0.492** | **0.337** |

Table 13: **Ablations** for Reference Scoring.

| Methods | Mean (RobustNeRF) | | |
|---|---|---|---|
| | PSNR↑ | SSIM↑ | LPIPS↓ |
| DGGS-TR ($N$=4) | 21.02 | 0.738 | 0.242 |
| DGGS-TR ($N$=8) | 20.22 | 0.690 | 0.311 |
| DGGS w/o Pruning ($N$=4) | **21.47** | **0.749** | **0.242** |

### B.7 QUALITATIVE COMPARISON OF PREDICTED MASKS

As shown in Fig. 16, our comparison between generalizable mask predictions, Robust Masks, and scene-specific trained masks reveals DGGS's capability to eliminate over-predicted regions from initial Robust Masks while achieving accurate results without scene-specific training. The predicted masks exhibit, unexpectedly, performance levels that rival scene-specific training approaches. Note that for fair evaluation, all masks are predicted under DGGS-TR without the involvement of any inference framework.

Our reference-based prediction proves effective for distractor mask estimation, an approach rarely addressed in previous distractor-free works. While NeRF-HuGS (Chen et al., 2024b) introduced similar concepts using SfM-based Heuristics, their approach relied heavily on predefined scene Colmap rather than direct reference inference, presenting substantial challenges in generalizable settings. Drawing inspiration from human perception, we observe that humans naturally identify transient objects (or distractor) across $N$ references from arbitrary scenes by leveraging 3D consistency to establish correspondences between different viewpoints. In this process, distractor typically manifest in regions exhibiting inconsistent correspondence relationships. Capitalizing on this fundamental insight, DGGS identifies non-distractor regions scene-agnostically through the exploitation of stable references residual loss and 3D consistency of static regions. This process, detailed in Sec. 4.1.1, effectively functions as a filtering mechanism for the Robust Masks based on references, establishing a novel paradigm for references-based distractor mask prediction.

### B.8 QUALITATIVE COMPARISON IN MORE SYNTHETIC SCENES

We report more results on synthetic scenes in Fig. 11. To avoid the impact of selecting different references, here we compare the qualitative results of Mvsplat* and DGGS-TR under the same references. It is not difficult to notice that Mvsplat* (Pretrain) is severely affected by distractor, especially with the presence of artifacts and blurring. This blurring even appears in some non-distractor areas, suggesting that the 3D inconsistencies might have influenced the overall image encoding before decoding. Our DGGS-TR effectively mitigates these issues under the same references, but some artifacts still exist in certain cases (in the third and fifth rows). Then introducing an additional inference framework effectively alleviates this problem.

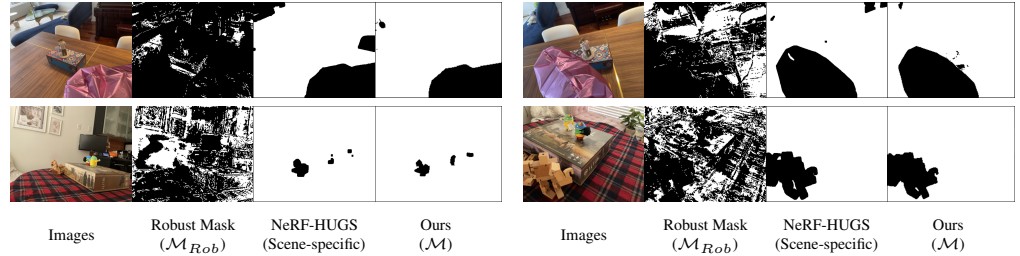

Figure 12: **Qualitative Comparison** for our scene-agnostic masks prediction vs. Robust Mask and Scene-specific training results.

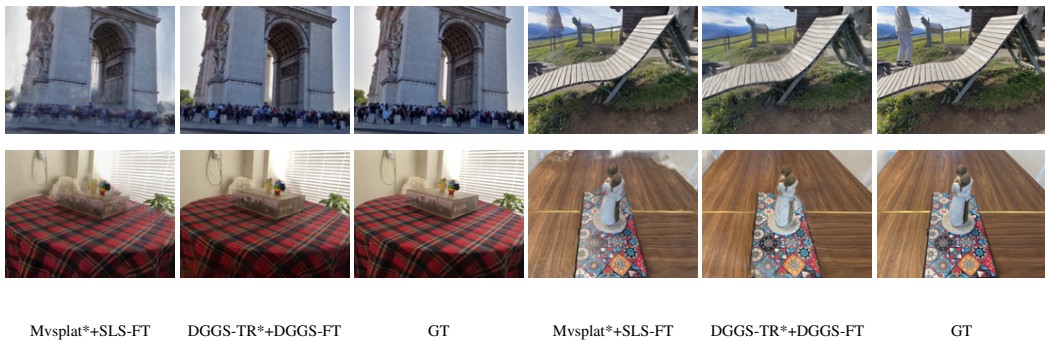

Figure 13: **Qualitative Experiments for the Fine-tuning Model.**

### B.9 QUALITATIVE COMPARISON OF FINE-TUNING MODEL

Fig. 13 reports the qualitative results of the fine-tuned model. Compared to the fine-tuning results of the SLS (Sabour et al., 2024) distractor masking method on the pre-trained Mvsplat*, DGGS-TR*+DGGS-FT demonstrates better results, which is consistent with Tab. 3. This is mainly because DGGS achieve more accurate distractor prediction, making the training process more stable.

### B.10 SINGLE-SCENE TRAINING SETTING

In addition to generalization capabilities, we investigate the performance of our reference-based paradigm in single-scene training configurations (Bao et al., 2023b).

Following distractor-free experimental protocols (Sabour et al., 2023; 2024; Chen et al., 2024a), Tab. 7 presents the performance of scene-specific trained DGGS-TR on the *statue* scene, trained on distractor-rich data ('clutter') without access to distractor-free references during inference 'extra'. Notably, DGGS achieves comparable results to existing approaches despite the absence of explicit iterative optimized representations. Moreover, when simulating practical scenarios by introducing partial distractor-free references during inference (while maintaining partial distractor-rich references, corresponding to the bottom row of the Tab. 7), DGGS exhibits superior performance. This performance can be primarily attributed to DGGS's accurate distractor prediction capabilities.

## C FAILURE CASES

Fig. 16 presents an analysis of failure cases. As mentioned in Sec. 6, although DGGS effectively mitigates distractor effects during both training and inference phases, it encounters difficulties with consistently occluded regions across reference views and novel areas. Examples include regions occluded by robots that are not visible in other views (left) and areas consistently blocked by a car in the lower-left corner (right).

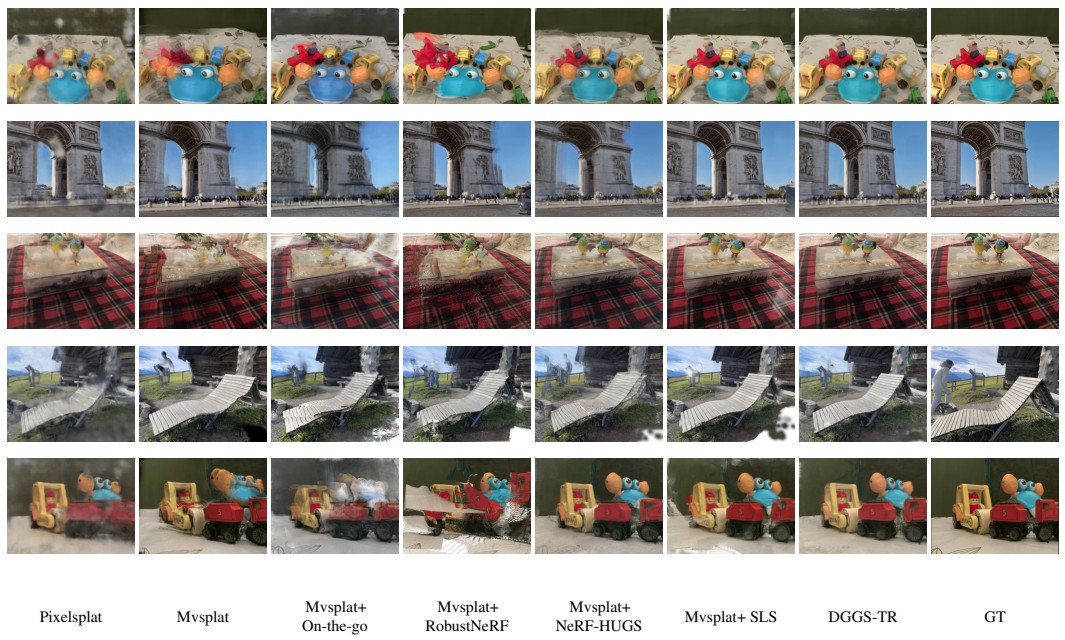

| Pixelsplat | Mvsplat | Mvsplat+
On-the-go | Mvsplat+
RobustNeRF | Mvsplat+
NeRF-HUGS | Mvsplat+ SLS | DGGS-TR | GT |

Figure 14: **More Qualitative Comparison of Re-trained Existing Methods** across unseen scenes.

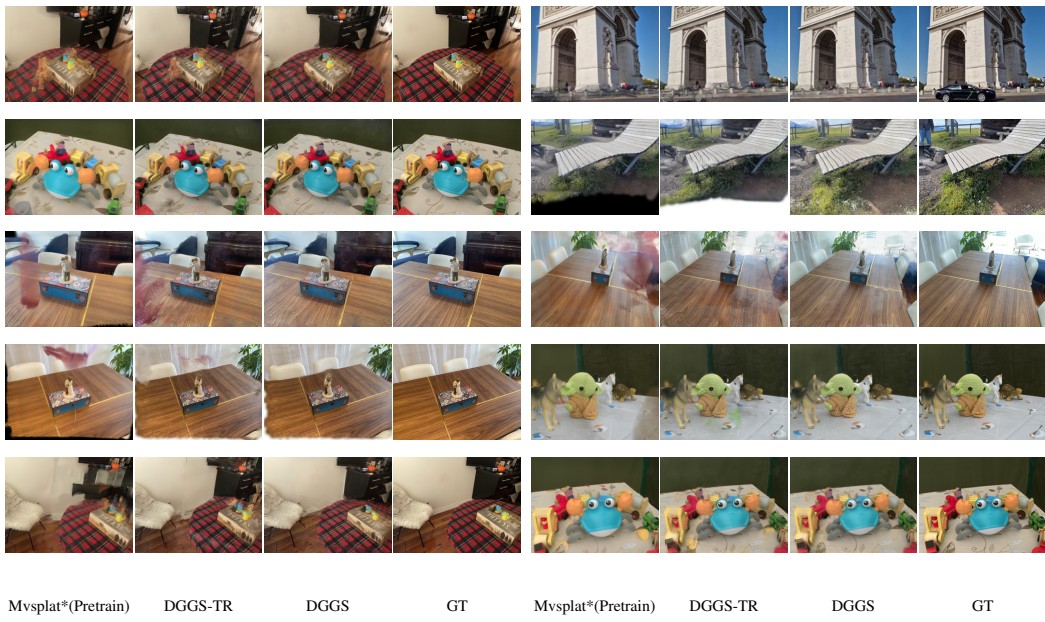

| Mvsplat*(Pretrain) | DGGS-TR | DGGS | GT | Mvsplat*(Pretrain) | DGGS-TR | DGGS | GT |

Figure 15: **More Qualitative Comparison of Pre-trained Models and our DGGS-TR as well as DGGS** under unseen scenes.

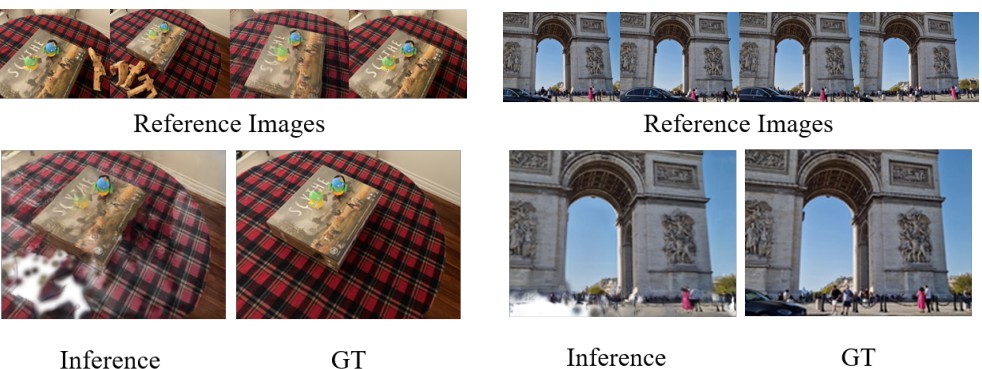

Figure 16: **Failure Cases**.

