# OpenReview forum: "Distractor-free Generalizable 3D Gaussian Splatting"
_ICLR.cc/2026/Conference — ICLR 2026 Poster_

### Official Review · Reviewer_ArTN · 2025-10-24

**Soundness:** 2
**Presentation:** 1
**Contribution:** 2
**Rating:** 4
**Confidence:** 4

**Summary:**

The authors propose Distractor-free Generalizable 3D Gaussian Splatting, a method designed for removing transient objects in feedforward GS methods. The main goal is to find proper transient masks and remove transients from optimizing the feedforward network. Firstly, DGGS implemented a robust mask based on robustnerf. Such mask is improved by incorporating a mask_ref (from re-rendered photometric loss) from reference images and project it on the target pose. Such mask is further improved based on multi-view visibility and combines with robustnerf. At test time, DGGS introduces schemes to reduce the usage of images with strong transient objects, and perform additional optimization to remove artifacts in the scene.

**Strengths:**

1. The authors demonstrated superior performances in feedforward 3DGS compared to prior work based on PSNR/SSIM etc.
3. The masks seem reasonable, and the resultant visual quality is also improved.

**Weaknesses:**

In general, I think the writing requires a lot of polishing, in terms of readability, background knowledge, e.g., segmentation models which play a significant role, and overall importance/novelty of the method.

After reading this work, the sense that I get is that, while performance is certainly good, the paper heavily leans on engineering and parameter tuning. This includes multiple important hyperparameters (threshold for masks), requirement for estimated depth to be reasonable for reprojection, integration of an external segmentation model that is minimally mentioned, test time hyperparameters (N views - BTW, N is defined multiple times with different meanings across the text). The overall insight is not very different from prior work, i.e., finding good transient masks such that rendering results can be improved by ignoring transients. I also have concerns about how this can be applied to larger scale scenes, as feedforward GS currently is relatively limited. Large scale scenes will lead to more noise in depth and multi-view inconsistencies, which this method seems to be very sensitive on.

I lean borderline on this work based on novelty; since there is no borderline, I lean towards borderline reject as I believe the writing can be better.

**Questions:**

1. The image resolution in the submitted PDF is really low. To the point where this is difficult to read.
2. Given that NeRF-HUGS produces similar masks as DGGS, can the authors expand on why mvsplat + NeRF-HUGS is significantly worse than DGGS?

---

> ### Author Response · Authors · 2025-11-20
> **Response ArTN(Part 1/3)**
>
> > **W1**: Segmentation models which play a significant role
>
> Thank you for raising this important point. Pre-trained entity segmentation has been widely used in previous scene-specific distractor-free studies, thus we adopt it in DGGS. However, **we would like to clarify that DGGS does not overly rely on pretrained segmentation models for 3DGS reconstruction.** It is mainly used to assist with an additional output: distractor masks. Next, we will further discuss and analyze from the perspectives of training and inference.
>
> ## Training:
> In the training process of the generalizable Distractor-free 3DGS task, we observe that the performance gain brought by the reference segmentation model is limited compared to other modules, as shown in the **fifth row of Tab. 2**. To further verify the role of the segmentation model in DGGS-TR, we further removed it from the mask refinement process, as shown in the supplementary table below.
>
> ## Inference:
> During inference, our Reference Scoring Mechanism also demonstrates a similar trend. **When using image segmentation at half resolution, the coarse segmentation results only lead to a PSNR drop of 0.05**. Thus, to improve efficiency, we propose incorporating segmentation results on low-resolution images, as discussed in **L-323 and L-522-524**. This demonstrates that DGGS has certain robustness to segmentation quality.
>
> To further validate this point, we have conducted additional tests on the inference performance after removing the pretrained segmentation model. Note that the training process here also does not use the pretrained segmentation model. Experiments demonstrate that compared to the improvement of DGGS over baseline and existing algorithms, the impact of the pretrained segmentation model on the results is limited.
>
> | Method | PSNR | SSIM | LPIPS |
> |--------|------|------|-------|
> | Baseline | 15.45 | 0.515 | 0.426 |
> | DGGS-TR | 21.02 | 0.738 | 0.242 |
> | DGGS-TR (w/o segmentation [newly added]) | 20.67 | 0.730 | 0.251 |
> | DGGS | 21.74 | 0.758 | 0.237 |
> | DGGS (w/o segmentation [newly added] based DGGS-TR (w/o segmentation)) | 21.41 | 0.749 | 0.240 |
>
> ## Analysis:
>
> During training stage, the core factors for stable generalizable 3DGS training are 3D consistency among references (see **Sec. 4.1**), not fine distractor segmentation boundaries. In other words, for some distractor regions that are misclassified as static objects without pre-trained segmentation, if their impact on 3D consistency is limited, then their impact on training is limited. In addition, thanks to multi-references inference and reference scoring mechanism, DGGS demonstrates a certain level of robustness in segmentation results during inference stage (see **Sec. 4.2**).
>
> *To further clarify to readers the role of the segmentation model in DGGS, we additionally supplement **Sec. 5.3.3 and Tab. 5** for analysis.*
>
> > **W2**. After reading this work, the sense that I get is that, while performance is certainly good, the paper heavily leans on engineering and parameter tuning. This includes multiple important hyperparameters (threshold for masks).
>
> We would like to clarify that, compared to traditional distractor-free algorithms [1] (which require adjusting multiple hyperparameters across different scenes), the core hyperparameter mentioned is only $\rho_{Ref}$, and this hyperparameter can remain consistent across any scene, as discussed in **Sec.A.2.1**. Therefore, we do not need to manually adjust hyperparameters for different scenes, which is also an advantage compared to previous scene-specific distractor-free works.
>
> Furthermore, we conducted an analysis of it in **Fig. 10**. As can be seen from the **Fig. 10**, different values of $\rho_{Ref}$ have some impact on the final results, but this impact is minor compared to the improvement our algorithm achieves over existing works.
>
> [1] RobustNeRF: Ignoring Distractor with Robust Losses
>
> > **W3**. Requirement for estimated depth to be reasonable for reprojection
>
> For existing feed-forward 3DGS related works, such as Mvsplat[2] and Depthsplat[3], these works often rely on these pretrained depth estimation models as the basis for projection and feature warping. Since DGGS aims to reduce the sensitivity of traditional generalizable 3DGS to distractors during training and inference, and is designed to be directly integrated into existing generalizable 3DGS frameworks, we adopt the same depth estimation setting as [2-3] to maintain compatibility and facilitate integration.
>
> [2] Mvsplat: Efficient 3d gaussian splatting from sparse multi-view images
>
> [3] Depthsplat: Connecting gaussian splatting and depth

---

> ### Author Response · Authors · 2025-11-20
> **Response ArTN(Part 2/3)**
>
> > **W4**. test time hyperparameters (*N* views)
>
> We would like to clarify that *N* and *K* are not testing hyperparameters, but rather **different experimental settings**, as discussed in [2-3]. In traditional generalizable 3DGS works [2-3], the network's performance under different inputs is discussed to demonstrate its practical capability. Therefore, we followed this setting and conducted discussion and analysis under different input experimental settings in **Fig. 10**, and we observed new trends in this DGGS task, as discussed in **Sec.5.3.2**.
>
> [2] Mvsplat: Efficient 3d gaussian splatting from sparse multi-view images
>
> [3] Depthsplat: Connecting gaussian splatting and depth
>
> *To prevent reader misunderstanding, we modify the **caption of Fig. 10**.*
>
> > **W5**. The overall insight is not very different from prior work, i.e., finding good transient masks such that rendering results can be improved by ignoring transients.
>
> We would like to clarify that the core insight of DGGS is to discuss how to predict distractor masks in **a feed-forward manner** and mitigate their impact during the generalizable 3DGS inference process under cross-scene setting. All traditional distractor-free related works rely on **iterative optimization** over a large number of same scene data, whereas our DGGS training paradigm can directly infer distractor **from limited references using 3D consistency in a feed-forward manner**. This is one of the core insights of this paper.
>
> Furthermore, traditional generalizable 3DGS methods directly predict 3DGS from reference pixels, which leads to distractor artifacts, as shown in the **first column of Fig. 7**. This means that even if we have accurate masks during inference, these methods still cannot produce clean 3DGS representations. Therefore, in addition to our proposed training paradigm, we also designed a new inference paradigm to effectively remove the artifacts caused by distractor. This is also fundamentally different from traditional distractor-free works that require per-scene iterative optimization, as they do not need to address the impact of distractor during inference. The detailed differences are shown in the table below.
>
> | Method | Training | Distractor Mask Prediction | Inference |
> |--------|-------|---------------------------|-----------|
> | Existing works | single-scene | Iterative Optimization | -|
> | Our DGGS | cross-scene | Feed-forward predict by 3D consistency | Distractor-free Inference Paradigm for 3DGS|
>
>
> > **W6**. I also have concerns about how this can be applied to larger scale scenes, as feedforward GS currently is relatively limited. Large scale scenes will lead to more noise in depth and multi-view inconsistencies, which this method seems to be very sensitive on.
>
> Thank you for raising this important point. We would like to clarify that DGGS is the first work focusing on mitigating the challenge of feed-forward GS in handling scenes with distractor. Therefore, similar to the baseline model (Mvsplat), to some extent it struggles to address the issues brought by large-scale scenes. However, the our training paradigm and inference paradigm of DGGS can be directly extended to other generalizable 3DGS related works, such as MVSplat360 [4] and Depthsplat [3], which is expected to further alleviate this issue in large-scale scenes and depth noise. For noise in depth and multi-view inconsistencies, these are also challenges for feedforward 3DGS. DGGS attempts to mitigate these issues during the mask prediction and refinement process, as mentioned in **L-254-257** and **Sec.4.1.2**.
>
> [4] Mvsplat360: Feed-forward 360 scene synthesis from sparse views
>
> > **W7**. In general, I think the writing requires a lot of polishing, in terms of readability; *N* is defined multiple times with different meanings across the text.
>
> Thank you for your suggestion. We would like to clarify that *N* only represents the number of references in our work.

---

> ### Author Response · Authors · 2025-11-20
> **Response ArTN(Part 3/3)**
>
> ***Now, we would like to answer your remaining questions:***
>
> > **Q1**: The image resolution in the submitted PDF is really low. To the point where this is difficult to read.
>
> We sincerely apologize. This is because the original file exceeded ICLR's maximum file upload size, so the images were compressed. We have re-uploaded a new version.
>
> > **Q2**: Given that NeRF-HUGS produces similar masks as DGGS, can the authors expand on why mvsplat + NeRF-HUGS is significantly worse than DGGS?
>
> It needs to be clarified that in **Fig. 6 and Fig. 12**, the NeRF-HUGS masks we compared against are **trained scene-by-scene(over 1h)**, while DGGS performs **feed-forward inference(0.148s)**. NeRF-HUGS does not have the capability to perform feed-forward inference of distractor masks, whereas DGGS can achieve feed-forward inference of distractor masks, and its prediction performance is even superior to NeRF-HUGS which requires scene-by-scene training. This further demonstrates the accuracy of our method for mask prediction. **However, if NeRF-HUGS is introduced into a generalizable setting, i.e., mvsplat + NeRF-HUGS, its mask prediction and reconstruction capability will severely degrade.**

---

> > ### Author Response · Authors · 2025-11-27
> > **Follow up**
> >
> > Dear Reviewer ArTN,
> >
> > We appreciate your time and effort in reading our response and revision! We have now provided detailed responses to all comments and made corresponding revisions where possible. With the discussion period ending around December 3, we would be very grateful if you could kindly take a moment to review our clarifications and let us know if any further information is needed.
> >
> > Best regards,
> >
> > Paper 17402 Authors

---

### Official Review · Reviewer_QPBd · 2025-10-28

**Soundness:** 2
**Presentation:** 3
**Contribution:** 2
**Rating:** 4
**Confidence:** 4

**Summary:**

The paper proposes DGGS, a framework designed to enable generalizable 3D Gaussian Splatting under distractor-rich real-world scenarios. Unlike existing generalizable 3DGS methods that assume clean static scenes, this work introduces: 1) A Reference-based Mask Prediction mechanism leveraging multi-view consistency. 2) A Mask Refinement module using segmentation priors and occlusion-aware auxiliary supervision. 3) A Two-stage inference strategy, including reference scoring and 3D gaussian primitive pruning, to suppress inference-time artifacts.

**Strengths:**

1. The use of multi-view geometric consistency to correct masks, reflects good insight rather than brute force.

2. Comprehensive experiments, including synthetic distractor construction.

3. Inference-time pruning is practical and effective.

**Weaknesses:**

1. Heavy reliance on segmentation priors. The pipeline is not truly “feed-forward generalizable” if high-quality segmentation is required and pre-computed.

2. Reference stability assumption unproven. The paper does not quantify how often reference re-rendering is accurate enough to serve as a stable supervisory source.

3. Efficiency cost. Two-stage inference + segmentation noticeably sacrifices speed, which is a key appeal of 3DGS.

4. Mask failure modes not fully analyzed. The limitations section mentions occlusions, but no systematic characterization is provided.

**Questions:**

1. How robust is the reference mask filtering when references also contain distractors?

2. What is the computational overhead of the full pipeline?

3. Does the segmentation model need retraining or domain adaptation in unseen categories?

4. Could the mask refinement be done without segmentation, e.g., self-supervised feature aggregation?

5. How does performance degrade with increasing viewpoint disparity among references?

---

> ### Author Response · Authors · 2025-11-20
> **Response QPBd(Part 1/3)**
>
> Thank you for taking the time to review our work and provide detailed feedback! We greatly appreciate your recognition of the good insight regarding the use of multi-view geometric consistency to correct masks and the effectiveness of our proposed method.
>
> *We also noticed you have some constructive questions about our work, and we're happy to elaborate further below!*
>
> > **W1**. Heavy reliance on segmentation priors. The pipeline is not truly “feed-forward generalizable” if high-quality segmentation is required and pre-computed.
>
> Thank you for raising this important point. Pre-trained entity segmentation has been widely used in previous scene-specific distractor-free studies, thus we adopt it in DGGS. **However, we would like to clarify that DGGS does not overly rely on pretrained segmentation models for 3DGS reconstruction.** It is mainly used to assist with an additional output: distractor masks. Next, we will further discuss and analyze from the perspectives of training and inference.
>
> ## Training:
> In the training process of the generalizable Distractor-free 3DGS task, we observe that the performance gain brought by the reference segmentation model is limited compared to other modules, as shown in the **fifth row of Tab. 2**. To further verify the role of the segmentation model in DGGS-TR, we further removed it from the mask refinement process, as shown in the supplementary table below.
>
> ## Inference:
>
> During inference, our Reference Scoring Mechanism also demonstrates a similar trend. **When using image segmentation at half resolution, the coarse segmentation results only lead to a PSNR drop of 0.05.** Thus, to improve efficiency, we propose incorporating segmentation results on low-resolution images, as discussed in **L-323 and L-522-524**. This demonstrates that DGGS has certain robustness to segmentation quality.
>
> To further validate this point, we have conducted additional tests on the inference performance after removing the pretrained segmentation model. Note that the training process here also does not use the pretrained segmentation model. Experiments demonstrate that compared to the improvement of DGGS over baseline and existing algorithms, the impact of the pretrained segmentation model on the results is limited.
>
> | Method | PSNR | SSIM | LPIPS |
> |--------|------|------|-------|
> | Baseline | 15.45 | 0.515 | 0.426 |
> | DGGS-TR | 21.02 | 0.738 | 0.242 |
> | DGGS-TR (w/o segmentation [newly added]) | 20.67 | 0.730 | 0.251 |
> | DGGS | 21.74 | 0.758 | 0.237 |
> | DGGS (w/o segmentation [newly added] based DGGS-TR (w/o segmentation)) | 21.41 | 0.749 | 0.240 |
>
> ## Analysis:
>
> During training stage, the core factors for stable generalizable 3DGS training are 3D consistency among references (see **Sec. 4.1**), not fine distractor segmentation boundaries. In other words, for some distractor regions that are misclassified as static objects without pre-trained segmentation, if their impact on 3D consistency is limited, then their impact on training is limited. In addition, thanks to multi-references inference and reference scoring mechanism, DGGS demonstrates a certain level of robustness in segmentation results during inference stage (see **Sec. 4.2**).
>
> *To further clarify to readers the role of the segmentation model in DGGS, we additionally supplement **Sec. 5.3.3 and Tab. 5** for analysis.*

---

> ### Author Response · Authors · 2025-11-20
> **Response QPBd (Part 2/3)**
>
> > **W2**. Reference stability assumption unproven. The paper does not quantify how often reference re-rendering is accurate enough to serve as a stable supervisory source.
>
> Thank you for your suggestion. For most generalizable 3DGS methods, the performance of reference re-rendering is always more robust compared to other viewpoints, as discussed for the pretrained Mvsplat in the supplementary table below, yet this property has been rarely exploited in previous works.
>
> ## The Role of Reference Re-rendering:
> We need to clarify that reference re-rendering is not directly used to supervise the training process of DGGS; it serves as additional auxiliary guidance for the mask prediction of the query view, as discussed in **Sec.4.1.1**.
>
> ## Experiments
> We conducted additional experiments in the synthetic dataset to analyze the quality of reference re-rendering to demonstrate the correctness of our assumption related to reference re-rendering.
>
> | Method (PSNR) | Train Stage | Query view | Reference Re-rendering view |
> |--|-|-|-|
> | Trained Mvsplat | Done | 26.02 | 29.71 |
> | DGGS-TR | 200 | 12.13 | 17.82 (Mask) |
> | DGGS-TR | 1000 | 16.36 | 20.38 (Mask) |
> | DGGS-TR | 10000 | 22.11 | 25.65 (Mask) |
> | Trained DGGS-TR | Done| 26.51 | 28.02 (Mask) |
>
> From the experimental results, it is not difficult to observe that reference re-rendering exhibits better performance compared to the query view from the beginning of generalizable 3DGS training. This property enables reference re-rendering to be used to guide distractor mask prediction for the query view. As the training progresses, the performance gap between the two gradually narrows until the training concludes. Furthermore, for the pretrained Mvsplat, the same trend is also observed on the Re-10k dataset.
>
> ## Re-rendered reference mask visualization:
> To verify that Reference Re-rendering has the ability to identify distractor within it, we have visualized some of the reference masks in **Fig. 1**. DGGS can directly take references containing distractor as input and predict the corresponding reference masks and 3DGS representation through feed-forward inference.
>
> *To further demonstrate to readers the stability of reference re-rendering, we additionally supplement **Sec.5.2.2 and Tab. 6** for analysis.*
>
> > **W3.** Efficiency cost. Two-stage inference + segmentation noticeably sacrifices speed, which is a key appeal of 3DGS.
>
> **Inference speed is not Rendering Speed.**
>
> I need to clarify that the **inference speed mentioned in DGGS refers to the reconstruction speed of 3DGS, not the rendering speed**. Our two-stage inference + segmentation does not affect the rendering speed of 3DGS; it is only related to the inference speed of 3DGS (i.e., the process from images to reconstruct 3DGS). Compared to traditional distractor-free works that require over 1 hour of training time, our DGGS only needs 0.148s for feed-forward reconstruction. The detailed time analysis is as follows:
>
> | Method | Generalizable|Inference time| Rendering time |
> |-|-|-|-|
> |Traditional 3DGS|✗| over 1 hour (training) | 0.002s |
> |Traditional distractor-free 3DGS works | ✗ | over 1 hour (training) | 0.002s |
> |DGGS|✓|**0.148s** (feed-forward) | 0.002s |
>
> Therefore, it is not difficult to see from the above table that DGGS does not affect the important rendering speed of 3DGS, and compared to traditional distractor-free methods, it significantly reduces inference time.
>
> > **W4**. Mask failure modes not fully analyzed. The limitations section mentions occlusions, but no systematic characterization is provided.
>
> Thank you for raising this important point.
> ## Limitations:
> Due to limited space in the original main text, we discussed failure cases and visualized them in **Appendix C**. Since DGGS does not have additional generative capabilities, it will exhibit phenomena such as speckles when facing situations with common occlusions among all references. This issue can be mitigated by introducing generative models, which can be discussed in future work.
>
> ## Mask failure scenarios
> For mask failure scenarios, during the inference process, if distractor are not segmented out, it will lead to additional distractor artifacts in the 3DGS scene, similar to the results of Mvsplat* (Pretrain) in **Fig. 7**. On the other hand, if a large number of non-distractor regions are identified as distractor regions, it will cause some speckles to appear, as shown in **Fig. 16**. Notably, thanks to the reference-based mask prediction and refinement strategy proposed by DGGS, DGGS can directly perform feed-forward inference to obtain high-precision masks, even surpassing NeRF-HUGS which requires scene-by-scene training, as shown in **Fig. 6 and Fig. 12**. Furthermore, as we demonstrate in **Q1**, DGGS possesses certain robustness to inaccurate masks.
>
> *To further discuss the case of common occlusion, we expand the description in the Limitation section and provide association in Appendix C.*

---

> ### Author Response · Authors · 2025-11-20
> **Response QPBd(Part 3/3)**
>
> ***Now, we would like to answer your remaining questions:***
>
> > **Q1:** How robust is the reference mask filtering when references also contain distractor?
>
> I would like to clarify that, in fact, in all our experiments on real data (RobustNeRF and On-the-go) as well as synthetic datasets, the reference images contain a large number of distractor, as shown in **Fig. 1, 2, 3, 8, and 11**. It can be seen that DGGS can fully handle such scenarios, even when references contain a large number of distractor (even all references contain distractor). This is because DGGS primarily relies on the static regions of references to guide distractor prediction in the query view and relatively stable reference re-rendering.
>
> > **Q2**: What is the computational overhead of the full pipeline?
>
> During inference, under the original setting with *N*(*K*)=4(8) and an image size of 252, DGGS's memory usage is 10G. When the resolution increases to 504, DGGS's memory usage is 16G. During training, since DGGS can perform pre-segmentation, its computational overhead is essentially consistent with Mvsplat (baseline).
>
> *To further analyze memory usage and time, we conduct additional ablation experiments and analysis under different *N*(*K*) and different resolutions in **Sec. B.2 and Tab. 10.** If you have further questions or additional suggestions, we welcome continued discussion.*
>
> > **Q3**. Does the segmentation model need retraining or domain adaptation in unseen categories?
>
> In our experiments, we did not perform retraining or domain adaptation on the pretrained segmentation models, whether on real or synthetic data. As mentioned earlier, the performance of segmentation models has a limited impact on DGGS reconstruction, and the core of DGGS lies in how to use 3D consistency between references to determine the locations of distractor. Therefore, if the goal is solely reconstruction (rather than accurate distractor masks), we did not make additional retraining on the pre-trained segmentation models.
>
> > **Q4**. Could the mask refinement be done without segmentation, e.g., self-supervised feature aggregation?
>
> Thank you for your suggestion. This is indeed a feasible alternative that can replace the segmentation network with self-supervised feature aggregation. This is because the essential role of the segmentation model is to filter/complete misclassified parts, which can also be achieved by introducing clustering strategies and feature consistency terms. Its advantage lies in removing the additional segmentation model from the framework, but it would make it difficult to obtain accurate distractor masks as an additional output of this work. To verify this method, we conducted additional experiments.
>
> | Method | PSNR | SSIM | LPIPS |
> |-|-|-|-|
> | DGGS-TR (feature consistency terms) | 20.85 | 0.733 | 0.242 |
> | DGGS-TR (clustering) | 20.92 | 0.736 | 0.245 |
> | DGGS-TR (our)|21.02|0.738|0.242|
>
> *To further analyze the role of the segmentation model, we conduct additional ablation experiments (as mentioned above) and analysis in **Sec. B.4 and Tab. 11**. If you have further questions or additional suggestions, we welcome continued discussion.*
>
> > **Q5**. How does performance degrade with increasing viewpoint disparity among references?
>
> ## Theoretical Analysis
>
> As the reference viewpoint angle increases, the reconstruction performance of DGGS will relatively weaken (which is consistent with existing works, such as Mvsplat[1]). Additionally, its ability to identify distractor will decrease. This is because DGGS essentially relies on 3D consistency between reference images to infer the distractor position. Assuming there is no or little overlap between reference viewpoints, then the limited 3D consistent regions make it difficult to facilitate the prediction of distractor regions.
>
> ## Additional Experiments
>
> We additionally discussed a set of ablation experiments on increasing viewpoint disparity among references without re-training model.
>
> | Method | PSNR | SSIM | LPIPS |
> |--------|------|------|-------|
> | DGGS (increasing viewpoint disparity) | 20.99 | 0.739 | 0.245 |
> | DGGS (our) | 21.74 | 0.758 | 0.237 |
>
> In the experiments, we increased the viewpoint disparity of reference selection, and its performance showed a significant decline.
>
> [1] Mvsplat: Efficient 3d gaussian splatting from sparse multi-view images

---

> > ### Author Response · Authors · 2025-11-27
> > **Follow up**
> >
> > Dear Reviewer QPBd,
> >
> > We appreciate your time and effort in reading our response and revision! We have now provided detailed responses to all comments and made corresponding revisions where possible. With the discussion period ending around December 3, we would be very grateful if you could kindly take a moment to review our clarifications and let us know if any further information is needed.
> >
> > Best regards,
> >
> > Paper 17402 Authors

---

### Official Review · Reviewer_sapT · 2025-10-31

**Soundness:** 4
**Presentation:** 4
**Contribution:** 3
**Rating:** 6
**Confidence:** 3

**Summary:**

This paper proposes a framework called DGGS aimed at making generalizable 3D Gaussian Splatting (3DGS) robust against distractors (transient objects) in real-world scenes. Existing generalizable 3DGS models assume static environments and suffer from training instability and artifacts when transient objects appear. DGGS mitigates this by predicting distractor masks through multi-view geometric consistency and refining them with segmentation priors, then using these masks to exclude distractors during training. At inference, it selects cleaner reference views and prunes distractor-related Gaussian primitives. Experiments demostrate this method outperform the generalizable 3DGS baselines, even better than some scene-specific distractor removal techniques.

**Strengths:**

- This is the first work addressing distractors in generalizable 3DGS, filling an important gap in real-world usage.
- It significantly boosts robustness and reconstruction quality compared to both baseline 3DGS models and naively transferred scene-specific distractor-free methods.
- The approach generalizes well to unseen scenes and improves inference quality via smart reference selection and pruning.

**Weaknesses:**

- The method relies on several additional modules, but the sensitivity of the overall performance to the choice or quality of these modules is not discussed.
- The quality of the generated masks depends on the accuracy of segmentation and depth estimation, which may lead to failure cases in scenes with heavy occlusions or imprecise geometry.
- Since the approach depends on mask generation, it is unclear how well it would handle naturally dynamic environments, such as moving trees or water, where mask accuracy could be compromised.

**Questions:**

Out of curiosity, can the two-stage inference be made more efficient for real-time use? Have you tested foundation segmentation models like SAM-2, and do they meaningfully improve results?

---

> ### Author Response · Authors · 2025-11-20
> **Response sapT (Part 1/2)**
>
> Thank you for taking the time to review our work and provide detailed feedback! We greatly appreciate your recognition of the importance of the distractor-free generalizable 3DGS problem and the effectiveness of our proposed method.
>
> *We also noticed you have some constructive questions about our work, and we're happy to elaborate further below! For convenience, we will attempt to discuss some similar weaknesses and questions together, and then separately address the remaining questions afterward.*
>
> > **W1&Q2**. The method relies on several additional modules, but the sensitivity of the overall performance to the choice or quality of these modules is not discussed. Have you tested foundation segmentation models like SAM-2, and do they meaningfully improve results?
>
> Thank you for raising this important point. Compared to previous generalizable 3DGS works [1], DGGS mainly introduces additional segmentation models (Entity Segmentation Model), **which have already been applied in traditional scene-specific distractor-free works [2]**. We followed previous works [2] for the segmentation model, and this is not the main contribution of DGGS. Therefore, the paper does not provide much discussion on their selection.
>
> Furthermore, we would like to clarify that DGGS does not overly rely on pretrained segmentation models. In the training process of the generalizable Distractor-free 3DGS task, we observe that the performance gain brought by the reference segmentation model is limited compared to other modules, as shown in the **fifth row of Tab. 2**. To further verify the role of the segmentation model in DGGS-TR, we further removed it from the mask refinement process, as shown in the supplementary table below.
>
> During inference, our Reference Scoring Mechanism also demonstrates a similar trend. **When using image segmentation at half resolution, the coarse segmentation results only lead to a PSNR drop of 0.05.** Thus, to improve efficiency, we propose incorporating segmentation results on low-resolution images, as discussed in **L-323** and **L-522-524**. This demonstrates that DGGS has certain robustness to segmentation quality.
>
> To further validate this point, we have conducted additional tests on the inference performance after removing the pretrained segmentation model. Note that the training process here also does not use the pretrained segmentation model. Experiments demonstrate that compared to the improvement of DGGS over baseline (Mvsplat), the impact of the pretrained segmentation model on the results is limited.
>
> | Method | PSNR | SSIM | LPIPS |
> |--------|------|------|-------|
> |Baseline|15.45|0.515|0.426|
> | DGGS-TR| 21.02 | 0.738 | 0.242 |
> | DGGS-TR (w/o segmentation [newly added]) | 20.67 | 0.730 | 0.251 |
> | DGGS | 21.74 | 0.758 | 0.237 |
> | DGGS (w/o segmentation [newly added] based DGGS-TR (w/o segmentation)) | 21.41 | 0.749 | 0.240 |
>
> [1] Mvsplat: Efficient 3d gaussian splatting from sparse multi-view images
>
> [2] Entity-nerf: Detecting and removing moving entities in urban scenes
>
> ## SAM-2:
>
> To further demonstrate that our work is insensitive to different segmentation models and is robust to segmentation results, we conducted experiments by replacing the Entity Segmentation model with SAM2 (more standard) during both training and inference.
>
> | Method | PSNR | SSIM | LPIPS |
> |--------|------|------|-------|
> | DGGS-TR (SAM2 [newly added]) | 21.07 | 0.740 | 0.240 |
> | DGGS-TR (our) | 21.02 | 0.738 | 0.242 |
> | DGGS (SAM2 [newly added]) | 21.77 | 0.758 | 0.236 |
> | DGGS (our) | 21.74 | 0.758 | 0.237 |
>
> We found that although segmentation models like SAM-2 can predict more accurate distractor masks, the impact on the model's training and inference reconstruction results is limited.
>
> ## Analysis:
>
> During training stage, the core factors for stable generalizable 3DGS training are 3D consistency among references (see **Sec. 4.1**), not fine distractor segmentation boundaries. In other words, for some distractor regions that are misclassified as static objects without pre-trained segmentation, if their impact on 3D consistency is limited, then their impact on training is limited. In addition, thanks to multi-references inference and reference scoring mechanism, DGGS demonstrates a certain level of robustness in segmentation results during inference stage (see **Sec. 4.2**).
>
> *To further clarify to readers the role of the segmentation model in DGGS, we additionally supplement **Sec. 5.3.3 and Tab. 5** for analysis.*

---

> ### Author Response · Authors · 2025-11-20
> **Response sapT (Part 2/2)**
>
> > **W2.** The quality of the generated masks depends on the accuracy of segmentation and depth estimation, which may lead to failure cases in scenes with heavy occlusions or imprecise geometry.
>
> Thank you for raising this important point.
>
> ## Segmentation:
>
> As previously discussed in **W1**, if accurate additional output regarding distractor mask prediction is not pursued, compared to baseline, the impact of segmentation models on DGGS reconstruction training and inference is limited.
>
> ## Depth estimation:
>
> As for depth estimation, we adopted the pretrained UniMatch (following MVSplat[1]) as guidance to improve depth estimation accuracy. This paradigm (based on pretrained depth estimation models) has been widely applied in related works on generalizable 3DGS [1-2]. Furthermore, to mitigate masks predict errors caused by depth/warping, we introduced additional mask refinement to enhance the robustness of DGGS under different challenging scenarios, as discussed in **L-254-256 and Sec.4.1.2**. It is worth mentioning that the training paradigm proposed by DGGS can be introduced into other existing works such as Depthsplat[2], which incorporates the more accurate pre-trained Depth Anything V2 to further improve performance on depth/warping quality.
>
> Thanks to the pretrained depth estimation model, DGGS mitigates the impact of imprecise geometry. As for heavy occlusions, DGGS can still handle cases when a large number of distractor are present, such as the bus in **Fig. 1**. However, when all reference images have the common occlusions, DGGS may fail since it is a reconstruction model without generative capabilities, which we discussed in **Sec. Limitation and Fig.16**.
>
> [1] Mvsplat: Efficient 3d gaussian splatting from sparse multi-view images
>
> [2] Depthsplat: Connecting gaussian splatting and depth
>
> > **W3.** Since the approach depends on mask generation, it is unclear how well it would handle naturally dynamic environments, such as moving trees or water, where mask accuracy could be compromised.
>
> Consistent with previous scene specific distractor-free related works, DGGS mainly focuses on **mitigating the impact of transient objects (or distractor) on static scenes reconstruction [3-6]**. The distractor here are often dynamic, transient objects, as shown by the bus and balloons in **Fig. 1**. Moving trees or water cannot be considered as transiently appearing distractor; therefore, consistent with traditional scene specific distractor-free works, DGGS does not analyze such special cases. It is worth mentioning that if the focus is on naturally dynamic environments, such scenes should be reconstructed through generalizable 4DGS (or dynamic 3DGS), which is challenging for generalizable 3DGS.
>
> [3] Gaussian in the wild: 3d gaussian splatting for unconstrained image collections
>
> [4] Spotlesssplats: Ignoring distractor in 3d gaussian splatting.
>
> [5] Nerf on-thego: Exploiting uncertainty for distractor-free nerfs in the wild.
>
> [6] Nerf in the wild: Neural radiance fields for unconstrained photo collections.
>
> ***Now, we would like to answer your remaining questions:***
>
> > **Q1:** Out of curiosity, can the two-stage inference be made more efficient for real-time use?
>
> As mentioned in the paper, we attempted to improve the efficiency of two-stage inference by reducing the resolution of the first stage inference to half of the original, as mentioned in **L-323** and **L-522-524**. Of course, we can further compress inference time by further reducing the resolution in the first stage. Furthermore, if a certain degree of reconstruction performance sacrifice is acceptable, the two-stage inference can also be directly removed, using only Distractor Pruning during inference, and the results would still be far superior to existing works.

---

> > ### Author Response · Authors · 2025-11-27
> > **Follow up**
> >
> > Dear Reviewer sapT,
> >
> > We appreciate your time and effort in reading our response and revision! We have now provided detailed responses to all comments and made corresponding revisions where possible. With the discussion period ending around December 3, we would be very grateful if you could kindly take a moment to review our clarifications and let us know if any further information is needed.
> >
> > Best regards,
> >
> > Paper 17402 Authors

---

### Official Review · Reviewer_KDhL · 2025-11-01

**Soundness:** 3
**Presentation:** 3
**Contribution:** 3
**Rating:** 6
**Confidence:** 4

**Summary:**

This submission introduces DGGS, a new framework for Distractor-free Generalizable 3D Gaussian Splatting. It tackles two overlooked issues in generalizable 3DGS: (1) training instability due to transient distractors in real-world data, and (2) feed-forward inference artifacts caused by distractors in references. The method proposes a reference-based mask prediction that leverages 3D multi-view consistency to filter robust residual-based masks, a mask refinement stage that decouples disparity-induced errors and uses entity segmentation plus an auxiliary loss, and a two-stage inference procedure with reference scoring and distractor pruning. Extensive experiments on real (On-the-go, RobustNeRF) and synthetic data show consistent improvements over retrained generalizable baselines and scene-specific distractor-free approaches adapted to the generalizable setting, with additional gains from the inference stage.

**Strengths:**

- Clearly identifies and formulates a new, practically relevant problem: distractor-free generalizable 3DGS.

- Elegant reference-based mask filtering that reduces over-suppression typical of residual-only masks.

- Thoughtful mask refinement: decoupling disparity vs. distractor; auxiliary loss exploiting cross-view occlusion cues.

- Practical two-stage inference: reference scoring and 3D primitive pruning demonstrably reduce artifacts/holes.

- Strong empirical results with comprehensive comparisons, ablations, and both real and synthetic setups.

- Method is modular and can plug into existing generalizable 3DGS pipelines.

**Weaknesses:**

(1) Dependence on pre-trained entity segmentation during training and inference undermines full “feed-forward” purity and adds latency; domain robustness of the segmenter is not analyzed.

(2) The mask fusion strategy uses intersection across references (conservative), which may lead to under-coverage in low-overlap or high-parallax settings; the trade-off is not deeply quantified.

(3) Distractor pruning can introduce speckle/holes in commonly occluded areas; mitigation is heuristic and the failure modes are only briefly discussed.

(4) Some reliance on depth/warping quality from inferred 3DGS; failure cases when depth is noisy or textures are repeated are not thoroughly dissected.

(5) Fairness concerns: scene-specific methods are adapted into a generalizable training loop but may not reflect their best practices (e.g., stronger per-scene optimization), making cross-paradigm comparisons tricky.

(6) Efficiency overhead from two-stage inference and segmentation is non-trivial; the paper reports times but not detailed profiling or memory usage under varied K, N, and resolution.

**Questions:**

(1) How sensitive is performance to the quality of the pre-trained segmentation model and its domain shift (e.g., indoor vs. outdoor, low light)? Can lighter/zero-shot segmenters maintain most gains?

(2) Why choose strict intersection for multi-view mask fusion? Have you tried soft/weighted fusion (e.g., confidence weighting by photometric residuals or view angle) to recover more static pixels without raising distractor leakage?

(3) Can the auxiliary loss be extended with photometric/feature consistency terms to lessen dependence on segmentation?

(4) How robust is the approach when the majority of references contain similar distractors (e.g., many frames with the same moving car)? Does reference scoring still find sufficiently clean views?

(5) For pruning, did you evaluate per-primitive confidence aggregation across references (e.g., voting) instead of binary masking per view to reduce speckle?

(6) Could you report memory/time breakdown across stages (feature projection, mask prediction/refinement, scoring, pruning) and how they scale with K and N?

(7) Are there benefits or risks in training with the scoring mechanism online (curriculum-style selection of “cleaner” references) rather than only at inference?

(8) How does DGGS perform when camera intrinsics vary or are noisy? Is U assumed known and consistent?

---

> ### Author Response · Authors · 2025-11-20
> **Response KDhL (Part 1/5)**
>
> Thank you for taking the time to review our work and provide detailed feedback! We greatly appreciate your recognition of the importance of the distractor-free generalizable 3DGS problem and the effectiveness of our proposed method.
>
> *We also noticed you have some constructive questions about our work, and we're happy to elaborate further below! For convenience, we will attempt to discuss some similar weaknesses and questions together, and then separately address the remaining questions afterward.*
>
> > **W1&Q1**: W1 Dependence on pre-trained entity segmentation during training and inference undermines full “feed-forward” purity and adds latency; domain robustness of the segmenter is not analyzed. Q1 How sensitive is performance to the quality of the pre-trained segmentation model and its domain shift (e.g., indoor vs. outdoor, low light)? Can lighter/zero-shot segmenters maintain most gains?
>
> Thank you for raising this important point. Pre-trained entity segmentation has been widely used in previous scene-specific distractor-free studies, thus we adopt it in DGGS. **However, we would like to clarify that DGGS does not overly rely on pretrained segmentation models for 3DGS reconstruction.**
> ## Training:
> In the training process of the generalizable Distractor-free 3DGS task, we observe that the performance gain brought by the reference segmentation model is limited compared to other modules, as shown in the **fifth row of Tab. 2.** To further verify the role of the segmentation model in DGGS-TR, we further removed it from the mask refinement process, as shown in the supplementary table below.
> ## Inference:
> During inference, our Reference Scoring Mechanism also demonstrates a similar trend. **When using image segmentation at half resolution, the coarse segmentation results only lead to a PSNR drop of 0.05.** Thus, to improve efficiency, we propose incorporating segmentation results on low-resolution images, as discussed in **L-323 and L-522-524**. This demonstrates that DGGS has certain robustness to segmentation quality.
> To further validate this point, we have conducted additional tests on the inference performance after removing the pretrained segmentation model. Note that the training process here also does not use the pretrained segmentation model. Experiments demonstrate that compared to the improvement of DGGS over baseline (Mvsplat), the impact of the pretrained segmentation model on the results is limited.
>
> |Method|PSNR|SSIM|LPIPS|
> |-|-|-|-|
> | Baseline|15.45|0.515|0.426|
> | DGGS-TR|21.02|0.738|0.242|
> | DGGS-TR (w/o segmentation [**newly added**])|20.67|0.730|0.251|
> | DGGS|21.74|0.758|0.237|
> | DGGS (w/o segmentation [**newly added**] based DGGS-TR (w/o segmentation))|21.41|0.749|0.240|
>
> ## Analysis:
> During training stage, the core factors for stable generalizable 3DGS training are 3D consistency among references (see **Sec. 4.1**), not fine distractor segmentation boundaries. In other words, for some distractor regions that are misclassified as static objects without pre-trained segmentation, if their impact on 3D consistency is limited, then their impact on training is limited. In addition, thanks to multi-references inference and reference scoring mechanism, DGGS demonstrates a certain level of robustness in segmentation results during inference stage (see **Sec. 4.2**).
>
> ## Domain Robustness:
> Given that the improvement from pretrained segmentation models to DGGS is limited, the pretrained segmentation models used in DGGS are not fine-tuned on our real and synthetic datasets at all. In the experiments, we separately validated a large number of indoor (RobustNeRF), outdoor (On-the-go), and synthetic datasets. We found that although the performance of segmentation models varies, the impact on the model's training and inference reconstruction results is limited. Further illustrating that our method is relatively robust to erroneous segmentation results.
>
> To further demonstrate that our work is insensitive to different segmentation models and is robust to segmentation results, we added experiments by replacing the Entity Segmentation model with SAM2 (more standard) during both the training and inference.
>
> | Method |PSNR|SSIM|LPIPS|
> |-|-|-|-|
> |DGGS-TR (SAM2 [**newly added**]) |21.07|0.740|0.240|
> |DGGS-TR|21.02|0.738|0.242|
> |DGGS (SAM2[**newly added**]) |21.77|0.758|0.236|
> |DGGS|21.74|0.758|0.237|
>
> ## Lighter/zero-shot segmenters:
> For some zero-shot segmenters, they may bring certain improvements in distractor segmentation performance in more challenging scenarios, especially in cases with severe domain shift, while some more lightweight segmenters can also further improve efficiency, as we mentioned in the limitations. However, overall, they may be effective for segmentation distractor results, but their impact on reconstruction performance is limited.
>
> *To further clarify to readers the role of the segmentation model in DGGS, we additionally supplement **Sec. 5.3.3 and Tab. 5** in revision for analysis.*

---

> ### Author Response · Authors · 2025-11-20
> **Response KDhL (Part 2/5)**
>
> > **W2&Q2**: W2 The mask fusion strategy uses intersection across references, which may lead to under-coverage in low-overlap or high-parallax settings; the trade-off is not deeply quantified. Q2 Why choose strict intersection for multi-view mask fusion? Have you tried soft/weighted fusion (e.g., confidence weighting by photometric residuals or view angle) to recover more static pixels without raising distractor leakage?
>
> Thank you for raising this important point. Next, we will discuss from both training and inference perspectives why we chose the intersection strategy to fuse masks.
>
> ## Training:
> During the training process, we found that misclassifying distractor regions as non-distractor regions is more harmful than misclassifying non-distractor regions as distractor regions, as discussed in **L-253-255**. This is because incorrectly classifying distractor regions as non-distractor regions undermines the 3D consistency of the training data, which is important for the existing generalizable 3DGS paradigm. Furthermore, considering that there is noise in the prediction of reference masks, DGGS adopts a more conservative strategy (intersection) to mitigate the impact of distractor on the training process. For scenes with low-overlap or high-parallax settings, we discussed how to decouple disparity-induced errors in **Sec. 4.1.2**.
>
> | Method | PSNR | SSIM | LPIPS |
> |-|-|-|-|
> | DGGS-TR (50%) | 20.70 | 0.734 | 0.250 |
> | DGGS-TR| 21.02 | 0.738 | 0.242 |
>
> To further quantitatively verify this, we modified the original intersection-based mask fusion strategy to require at least 50% of the references to agree, meaning that if two references indicate that the corresponding location is a static region, it can be considered static. The experimental results show that this leads to a decline in the training model's performance, which may be due to distractor regions being misclassified as static regions, consistent with our previous analysis.
>
> ## Inference:
> During the inference process, this is indeed a trade-off, and we attempted to mitigate our conservative mask fusion strategy’s negative impact in the pruning strategy, as described in **Sec. 4.2.2**. As you mentioned, our approach may misclassify some static pixels as dynamic region masks, which leads to additional speckles as shown in **Fig. 16** (failure cases). Conversely, if dynamic regions are classified as static, this will inevitably result in additional artifacts, such as in the first column of **Fig. 7**. Therefore, we maintained a mask fusion strategy consistent with the training process in DGGS. To mitigate the issue of speckles (over-predicted distractor) caused by over-predicting distractor, we correspondingly modified the pruning strategy to prevent excessive pruning for a common occluded region for references, as shown in **L-353-356**.
>
> ## Soft/weighted fusion
> To discuss this issue, we conducted an analysis of different soft/weighted fusion approaches.
> **Photometric Residuals**: Regarding the choice of this strategy, since our masks are inferred from photometric residuals, using them as additional weights to aggregate masks may not yield the desired effectiveness. **View Angle**: Distractor masks are view-independent. This means that if a distractor appears in any reference view, regardless of how far or close it is, it will negatively affect the reconstruction quality of the entire scene. For view angle, we specifically studied the impact caused by decoupling disparity errors in **Sec. 4.1.2.** **To further validate this point, we additionally conducted ablation experiments, where we designed different weights based on different view angles (with higher weights for closer views) to assist mask fusion.**
>
> | Method | PSNR | SSIM | LPIPS |
> |-|-|-|-|
> | DGGS (view angles) | 21.60 | 0.752 | 0.240 |
> | DGGS| 21.74 | 0.758 | 0.237 |
>
> Experiments demonstrate that soft/weighted mask fusion based on view angles will lead to additional distractor artifacts in some cases, thereby reducing the quality of 3DGS reconstruction.
>
> *To further demonstrate the effectiveness of intersection fusion to readers, we conduct additional ablation experiments and analysis in **Sec. B.3 and Tab. 9**. If you have further questions or additional suggestions, we welcome continued discussion.*

---

> ### Author Response · Authors · 2025-11-20
> **Response KDhL (Part 3/5)**
>
> > **W3**: Distractor pruning can introduce speckle/holes in commonly occluded areas; mitigation is heuristic and the failure modes are only briefly discussed.
>
> We would like to clarify that we visualized and discussed failure cases in the appendix, especially commonly occluded areas, as shown in **Fig. 16**. Due to the lack of additional generative capabilities in DGGS, common occlusions are difficult to fix solely through reconstruction models from a limited number of references.
>
> As **L-354-356**, directly preventing the pruning of common regions across all reference views is a potential mitigation measure. DGGS is the first work to discuss distractor-free generalizable reconstruction. This issue can be further alleviated by additionally introducing generative models in future work, as mentioned in **L-522-524.**
>
> *To further discuss the case of common occlusion, we expand the description in the Limitation section and provide association in Appendix C.*
>
> > **W4**: Some reliance on depth/warping quality from inferred 3DGS; failure cases when depth is noisy or textures are repeated are not thoroughly dissected.
>
> Thank you for raising this important point. To mitigate issues caused by depth/warping quality, existing generalizable Mvsplat[1] introduced pretrained UniMatch as guidance to alleviate this problem (it also involves feature warping in Mvsplat), and our DGGS similarly follows this strategy. Since DGGS primarily focuses on discussing how to improve the robustness of generalizable 3DGS against distractor during training and inference, we did not discuss depth/warping failure cases, which theoretically align with our baseline (Mvsplat[1]). Furthermore, to mitigate masks predict errors caused by depth/warping, we introduced additional mask refinement to enhance the robustness of DGGS under challenging scenarios, as discussed in **L-254-256 and Sec.4.1.2**. It is worth mentioning that the training paradigm proposed by DGGS can be introduced into other existing works such as Depthsplat[2], which incorporates the more accurate pre-trained Depth Anything V2 to further improve performance on depth/warping quality.
>
> [1] Mvsplat: Efficient 3d gaussian splatting from sparse multi-view images
>
> [2] Depthsplat: Connecting gaussian splatting and depth
>
> > **W5**. Fairness concerns: scene-specific methods are adapted into a generalizable training loop but may not reflect their best practices (e.g., stronger per-scene optimization), making cross-paradigm comparisons tricky.
>
> Thank you for raising this important point. We would like to clarify that although most experiments were conducted under generalizable settings (DGGS mainly focuses on this setting.), we compared the fine-tuning results under single scene setting in **Tab. 3**. It is not difficult to see that the fine-tuning results of DGGS on single scenes are even better than the results of SpotLessSplats (as reported in SpotLessSplats[3]) trained on single scenes from scratch.
>
> | Method | PSNR | SSIM | LPIPS |
> |--------|------|------|-------|
> | SLS [newly added] | 22.53 | 0.77 | 0.18 |
> | DGGS-TR* + DGGS-FT | 23.85 | 0.787 | 0.128 |
>
> This benefits from DGGS's pretraining on large-scale datasets and its reference-based strategy for distractor regions prediction.
>
> Furthermore, since DGGS is the first work to study distractor-free generalizable 3DGS, we mainly focus on cross-scene task settings. Extensive experimental results demonstrate that existing scene-specific methods struggle to address this problem and highlight the importance of DGGS.
>
> [3] SpotlessSplats: Ignoring Distractor in 3D Gaussian Splatting
>
> *Correspondingly, we present the single-scene training results of SpotLessSplats in **Tab. 3** to further demonstrate the effectiveness of DGGS.*

---

> ### Author Response · Authors · 2025-11-20
> **Response KDhL (Part 4/5)**
>
> > **W6&Q6**. W6 Efficiency overhead from two-stage inference and segmentation is non-trivial; the paper reports times but not detailed profiling or memory usage under varied *K*, *N*, and resolution. Q6 Could you report memory/time breakdown across stages (feature projection, mask prediction/refinement, scoring, pruning) and how they scale with *K* and *N*?
>
> Thank you for raising this important point.
>
> ## Memory usage and time under varied *K*, *N*, and resolution:
>
> Thank you for your suggestion. We have additionally reported the efficiency under different *N*(*K*) and different resolutions, as shown in the table below. The impact of these factors on efficiency is similar to works like Mvsplat and Depthsplat, that is, as the resolution and *N* increase, the inference time and memory usage of DGGS also increases accordingly. Furthermore, as shown in **L-321-323**, the two-stage inference does not incur additional memory usage since it maintains the same number of references across the two stages.
>
> | *N* (*K*) | Resolution | Memory | Time (w/o seg) |
> |--|---|--|-|
> | 4 (8) | 252 | 10G | 0.111s |
> | 4 (8) | 504 | 16G | 0.277s |
> | 6 (12) | 252 | 12G | 0.203s |
>
> ## Time breakdown across stages:
>
> For the inference stage of DGGS, the efficiency cost is mainly related to the two-stage inference and the segmentation model, while the time consumption of other processes is often negligible, as discussed in **L-449**. Therefore, we mainly focus our analysis on the efficiency issues of these parts.
>
> | Stages | Time |
> |-|-|
> | Baseline | 0.084s |
> | Segmentation | $\approx$0.037s |
> | Two-stage inference | $\approx$0.027s |
> | DGGS | 0.148s |
>
> Thanks to the reduced reference resolution in the first inference stage, the additional inference time required by the two-stage inference is limited, as discussed in **L-323**. Furthermore, if accurate segmentation masks of distractor are not necessary, we can omit the segmentation module, which has limited impact on DGGS's reconstruction performance as **W1&Q1**, thereby further accelerating DGGS.
>
> *To further analyze memory usage and time, we conduct additional ablation experiments and analysis under different *N*(*K*) and different resolutions in **Sec. B.2 and Tab. 10**. If you have further questions or additional suggestions, we welcome continued discussion.*
>
>
> ***Now, we would like to answer your remaining questions:***
>
> > **Q3**:  Can the auxiliary loss be extended with photometric/feature consistency terms to lessen dependence on segmentation?
>
> ## Segmentation Role
> We would like to clarify that, DGGS primarily relies on 3D consistency between references to determine whether distractor position. In comparison, the introduction of segmentation models has a limited impact on the training process of DGGS as shown in **Tab.2** and **W1&Q1**.
>
> ## Photometric/feature consistency terms
>
> The main role of segmentation models during the training process is to complete/filter discrete points that are misclassified. In theory, they can be replaced by some clustering strategy and feature consistency terms, but similar to previous conclusions, their impact on model reconstruction training is limited, as shown in the following supplementary experiments. The difference is that these methods struggle to additionally output accurate distractor masks, while DGGS is able to achieve distractor mask prediction simultaneously during reconstruction, as shown in **Fig.1**.
>
> | Method | PSNR | SSIM | LPIPS |
> |-|-|-|-|
> | DGGS-TR (feature consistency terms) | 20.85 | 0.733 | 0.242 |
> | DGGS-TR (clustering) | 20.92 | 0.736 | 0.245 |
> | DGGS-TR (our)|21.02|0.738|0.242|
>
> *To further analyze the role of the segmentation model, we conduct additional ablation experiments (as mentioned above) and analysis in **Sec. B.4 and Tab. 11**. If you have further questions or additional suggestions, we welcome continued discussion.*
>
> > **Q4**: How robust is the approach when the majority of references contain similar distractor (e.g., many frames with the same moving car)? Does reference scoring still find sufficiently clean views?
>
> We would like to clarify that in most of our experiments, the majority of references contain similar distractor, or even all references contain the same dynamic distractor. For example, the green bus and balloons in **Fig. 1** appears extensively in multiple references and occupies a large amount of space, and DGGS is able to handle this type of scene. Additionally, in RobustNeRF and synthetic datasets (we synthesized the same distractor in all references in **Fig.11**.), the distractor in most scenes are identical in all references and our DGGS can still handle such scenarios, as shown in **Fig. 5,7,8,9,11**.
>
> In theory, DGGS is not affected by the same distractor appearing in most references, because DGGS's prediction of distractor is mainly related to the 3D consistency between references. The same moving objects will also lead to 3D inconsistency between references.

---

> ### Author Response · Authors · 2025-11-20
> **Response KDhL (Part 5/5)**
>
> > **Q5**: For pruning, did you evaluate per-primitive confidence aggregation across references (e.g., voting) instead of binary masking per view to reduce speckle?
>
> Thank you for your suggestions. Using per-primitive confidence aggregation across references as a basis for pruning may not yield the desired effectiveness. For pruning, its goal is to remove the Gaussian primitive where distractor exist in each reference. For a particular distractor, if it exists in only one reference image, it may still introduce additional slight artifacts. In this case, it is difficult to mitigate through the confidence aggregation approach. Considering the appearance of speckles, we can optionally disable pruning when all references corresponding to a certain region are occluded, as shown in **L-353-356**.
>
>
> > **Q7**:  Are there benefits or risks in training with the scoring mechanism online (curriculum-style selection of “cleaner” references) rather than only at inference?
>
> Thank you for your suggestions.
> We believe this may be effective for training stability. However, an online scoring mechanism will significantly increase training time. In comparison, directly controlling the number of distractor in the training data (such as our synthetic data) in a curriculum learning manner may be more efficient. For example, introducing more clean references in the early stages of training.
>
> > **Q8**. How does DGGS perform when camera intrinsics vary or are noisy? Is U assumed known and consistent?
>
> Yes, in fact, we followed the settings of most generalizable 3DGS works[1-2], that is, maintaining the same camera intrinsics and assuming they are known. However, it is worth mentioning that studying variable camera intrinsics is meaningful, which may be discussed in future work.
>
> [1] Mvsplat: Efficient 3d gaussian splatting from sparse multi-view images
>
> [2] Depthsplat: Connecting gaussian splatting and depth

---

> > ### Author Response · Authors · 2025-11-27
> > **Follow up**
> >
> > Dear Reviewer KDhL,
> >
> > We appreciate your time and effort in reading our response and revision! We have now provided detailed responses to all comments and made corresponding revisions where possible. With the discussion period ending around December 3, we would be very grateful if you could kindly take a moment to review our clarifications and let us know if any further information is needed.
> >
> > Best regards,
> >
> > Paper 17402 Authors

---

> > > ### Comment · Reviewer_KDhL · 2025-11-28
> > >
> > > Appreciate the responses from the authors with experiments in detail. The rebuttal has solve my concerns like dependence on pre-trained model and the mask fusion strategy. I will raise my score.

---

> > > > ### Author Response · Authors · 2025-11-28
> > > >
> > > > We are delighted to hear that our response and the revisions to the manuscript have addressed your concerns, and we greatly appreciate your decision to raise your score. Thank you once again for your time and valuable feedback!

---

### Author Response · Authors · 2025-11-20
**General Response**

## **General Response**

We thank all reviewers for their thoughtfull and helpfull feedback!

We are pleased that the reviewers (**KDhL**, **sapT**) recognize the practical significance of the new task of distractor-free generalizable 3DGS:

- "Clearly identifies and formulates a new, practically relevant problem: distractor-free generalizable 3DGS." **KDhL**
- "This is the first work addressing distractors in generalizable 3DGS, filling an important gap in real-world usage." **sapT**

We are also pleased that the reviewers (**KDhL**, **sapT**, **QPBd**) consider our proposed method elegant and having good insight:

- "Elegant reference-based mask filtering that reduces over-suppression typical of residual-only masks." **KDhL**
- "Thoughtful mask refinement: decoupling disparity vs. distractor; auxiliary loss exploiting cross-view occlusion cues." **KDhL**
- "The approach generalizes well to unseen scenes and improves inference quality via smart reference selection and pruning." **sapT**
- "The use of multi-view geometric consistency to correct masks, reflects good insight rather than brute force." **QPBd**

We are also glad that all reviewers find our proposed method reasonable, practical, and effective:

- "Strong empirical results with comprehensive comparisons, ablations, and both real and synthetic setups." **KDhL**
- "It significantly boosts robustness and reconstruction quality compared to both baseline 3DGS models and naively transferred scene-specific distractor-free methods."  **sapT**
- "Inference-time pruning is practical and effective."  **QPBd**
- "The authors demonstrated superior performances in feedforward 3DGS compared to prior work based on PSNR/SSIM etc."  **ArTN**
- "The masks seem reasonable, and the resultant visual quality is also improved."  **ArTN**

## **Updates and new experiments**

We summarise the new experiments and discussions inspired by the reviewer's comments below. We plan to incorporate changes into the PDF shortly after discussion with reviewers.

- To further clarify to readers the role of the segmentation model in DGGS, we additionally supplement **Sec. 5.3.3 and Tab. 5** for analysis.

- We conduct additional ablation experiments and analysis in **Sec. B.3 and Tab. 9**. to further demonstrate the effectiveness of intersection fusion to readers.

- To further analyze memory usage and time, we conduct additional ablation experiments and analysis under different *N*(*K*) and different resolutions in **Sec. B.2 and Tab. 10**.

- We conduct additional ablation experiments and analysis in **Sec. B.4 and Tab. 11** for some alternatives to segmentation models.

- To further demonstrate to readers the stability of reference re-rendering, we additionally supplement **Sec.5.2.2 and Tab. 6** for analysis.

- We expand the description in the Limitation section and improve the image resolution.


We sincerely thank all reviewers again for their valuable suggestions, and we welcome further discussion at any time.

---

### Author Response · Authors · 2025-12-03
**Summary of the Rebuttal Period**

Dear AC,

We would like to briefly summarize the rebuttal process for your convenience.

The reviewers' feedback has been constructive. Reviewers highlighted that our novel task (distractor-free generalizable 3DGS) "has practical significance" (**KDhL, sapT**), the proposed idea is "elegant, smart, thoughtful and demonstrates good insight" (**KDhL, sapT, QPBd**), the method is "superior, practical, effective and reasonable" (**KDhL, sapT, QPBd, ArTN**) and the empirical results are "strong, robust, and high-quality" (**KDhL, sapT, QPBd, ArTN**). Among the four reviewers:

**One** reviewer engaged in the discussion:

- *Reviewer KDhL* (**Original Rating 6 / Confidence 4**): The rebuttal has solve my concerns like dependence on pre-trained model and the mask fusion strategy. **I will raise my score.**

**Three** reviewers did not have a chance to respond:

- *Reviewer sapT* (**Original Rating 6 / Confidence 3**)

- *Reviewer QPBd* (**Original Rating 4 / Confidence 4**)

- *Reviewer ArTN* (**Original Rating 4 / Confidence 4**)

For these reviewers, we've provided point-to-point clarifications and new results that strengthen the claims of the paper, including:

- Additional experiments to clarify that DGGS does **NOT** rely on segmentation models and is **robust** to the results of existing segmentation models in **Sec. 5.3.3 and Tab. 5**. (***Reviewer KDhL* acknowledged that this concern has been addressed.**)

- Provide quantitative analysis on reference re-rendering stability to validate our assumption in **Sec.5.2.2 and Tab. 6.**

- Clarified some potential **misunderstandings**[*concise response*] regarding :
  -  "**reliance on depth estimation**" [*following existing generalizable 3DGS works*],
  -  "**rendering efficiency cost**" [*does not affect rendering efficiency at all*],
  -  "**multiple important hyperparameters and parameter tuning**" [*only one hyperparameter with analysis provided and is scene-independent*],
  -  "**test-time hyperparameters**" [*They are different experimental settings following existing generalizable 3DGS works*],
  -  "**insight with existing per-scene optimization methods**" [*feed-forward mask prediction and two-stage feed-forward inference*],
  -  "**limitations on larger scale scenes**" [*This is a common problem for generalizable 3DGS; our DGGS can be plug-and-play integrated into other existing generalizable 3DGS works for large-scale scenes*].

- Move the detailed description and figures of limitations from the appendix to the main text.

We've updated the manuscript to incorporate the new results and discussions. Thank you for your time! We're happy to provide any further clarifications if needed.

Best regards,

Submission 17402 Authors

---

### Meta-Review · Area_Chair_pWew · 2025-12-19

**Summary:**

The paper proposes a new task aimed at mitigating distractors in generalizable 3DGS frameworks. The reviewers agree that the task is novel with clear practical significance, the proposed method is insightful and effective, and the experimental results are strong. The AC has carefully read the paper and agrees with these positive assessments.

During the rebuttal, the authors faithfully responded to the raised questions point by point. Since only Reviewer KDhL participated in the discussion, the AC carefully examined all responses and finds that most concerns have been well addressed, supported by additional experimental results provided during the rebuttal phase.

Overall, the AC believes that the strengths of this work outweigh the remaining weaknesses and is happy to recommend acceptance. When preparing the camera-ready version, the authors are encouraged to further polish the writing and presentation, particularly by improving the layout of sections (e.g., Sections 4.2.1 and 5.2, which currently contain limited content, could be merged into more appropriate places as regular paragraphs) and by refining figures and tables (e.g., enlarging some for better readability and moving some to the supplementary material where appropriate).

**Reviewer Concerns:**

Please refer to the summary above.

**Reviewer Scores:**

Reviewer KDhL: 6 (confirmed and would like to raise: "The rebuttal has solve my concerns like dependence on pre-trained model and the mask fusion strategy. I will raise my score.")
Reviewer sapT: 6 (no reply)
Reviewer QPBd: 4 (no reply)
Reviewer ArTN:4 (no reply)

---

### Decision · Program_Chairs · 2026-01-26

Accept (Poster)